# Convex and Non-convex Optimization Under Generalized Smoothness

**Haochuan Li**[*]
MIT
haochuan@mit.edu

**Jian Qian**[*]
MIT
jianqian@mit.edu

**Yi Tian**
MIT
yitian@mit.edu

**Alexander Rakhlin**
MIT
rakhlin@mit.edu

**Ali Jadbabaie**
MIT
jadbabai@mit.edu

## Abstract

Classical analysis of convex and non-convex optimization methods often requires the Lipschitz continuity of the gradient, which limits the analysis to functions bounded by quadratics. Recent work relaxed this requirement to a non-uniform smoothness condition with the Hessian norm bounded by an affine function of the gradient norm, and proved convergence in the non-convex setting via gradient clipping, assuming bounded noise. In this paper, we further generalize this non-uniform smoothness condition and develop a simple, yet powerful analysis technique that bounds the gradients along the trajectory, thereby leading to stronger results for both convex and non-convex optimization problems. In particular, we obtain the classical convergence rates for (stochastic) gradient descent and Nesterov's accelerated gradient method in the convex and/or non-convex setting under this general smoothness condition. The new analysis approach does not require gradient clipping and allows heavy-tailed noise with bounded variance in the stochastic setting.

## 1 Introduction

In this paper, we study the following *unconstrained* optimization problem

$$\min_{x \in \mathcal{X}} f(x), \tag{1}$$

where $\mathcal{X} \subseteq \mathbb{R}^d$ is the domain of $f$. Classical textbook analyses [Nemirovskij and Yudin, 1983, Nesterov, 2003] of (1) often require the Lipschitz smoothness condition, which assumes $\left\|\nabla^2 f(x)\right\| \leq L$ almost everywhere for some $L \geq 0$ called the smoothness constant. This condition, however, is rather restrictive and only satisfied by functions that are both upper and lower bounded by quadratic functions.

Recently, Zhang et al. [2019] proposed the more general $(L_0, L_1)$-smoothness condition, which assumes $\left\|\nabla^2 f(x)\right\| \leq L_0 + L_1 \left\|\nabla f(x)\right\|$ for some constants $L_0, L_1 \geq 0$, motivated by their extensive language model experiments. This notion generalizes the standard Lipschitz smoothness condition and also contains e.g. univariate polynomial and exponential functions. For *non-convex* and $(L_0, L_1)$-smooth functions, they prove convergence of gradient descent (GD) and stochastic gradient descent (SGD) *with gradient clipping* and also provide a complexity lower bound for *constant-stepsize* GD/SGD without clipping. Based on these results, they claim gradient clipping or other forms of adaptivity *provably* accelerate the convergence for $(L_0, L_1)$-smooth functions. Perhaps due to the

---

[*]Equal contribution.

37th Conference on Neural Information Processing Systems (NeurIPS 2023).

lower bound, all the follow-up works under this condition that we are aware of limit their analyses to adaptive methods. Most of these focus on non-convex functions. See Section 2 for more discussions of related works.

In this paper, we significantly generalize the $(L_0, L_1)$-smoothness condition to the $\ell$-smoothness condition which assumes $\left\|\nabla^2 f(x)\right\| \leq \ell(\|\nabla f(x)\|)$ for some non-decreasing continuous function $\ell$. We develop a simple, yet powerful approach, which allows us to obtain stronger results for *both convex and non-convex* optimization problems when $\ell$ is sub-quadratic (i.e., $\lim_{u\to\infty} \ell(u)/u^2 = 0$) or even more general. The $\ell$-smooth function class with a sub-quadratic $\ell$ also contains e.g. univariate rational and double exponential functions. In particular, we prove the convergence of *constant-stepsize* GD/SGD and Nesterov's accelerated gradient method (NAG) in the convex or non-convex settings. For each method and setting, we obtain the classical convergence rate, under a certain requirement of $\ell$. In addition, we relax the assumption of bounded noise to the weaker one of bounded variance with the simple SGD method. See Table 1 for a summary of our results and assumptions for each method and setting. At first glance, our results "contradict" the lower bounds on constant-stepsize GD/SGD in [Zhang et al., 2019, Wang et al., 2022]; this will be reconciled in Section 5.3.

Our approach analyzes boundedness of gradients along the optimization trajectory. The idea behind it can be informally illustrated by the following "circular" reasoning. On the one hand, if gradients along the trajectory are bounded by a constant $G$, then the Hessian norms are bounded by the constant $\ell(G)$. Informally speaking, we essentially have the standard Lipschitz smoothness condition[2] and can apply classical textbook analyses to prove convergence, which implies that gradients converge to zero. On the other hand, if gradients converge, they must be bounded, since any convergent sequence is bounded. In other words, the bounded gradient condition implies convergence, and convergence also implies the condition back, which forms a circular argument. If we can break this circularity of reasoning in a rigorous way, both the bounded gradient condition and convergence are proved. In this paper, we will show how to break the circularity using induction or contradiction arguments for different methods and settings in Sections 4 and 5. We note that the idea of bounding gradients can be applied to the analysis of other optimization methods, e.g., the concurrent work [Li et al., 2023] by subset of the authors, which uses a similar idea to obtain a rigorous and improved analysis of the Adam method [Kingma and Ba, 2014].

**Contributions.** In light of the above discussions, we summarize our main contributions as follows.

- We generalize the standard Lipschitz smoothness and also the $(L_0, L_1)$-smoothness condition to the $\ell$-smoothness condition, and develop a new approach for analyzing convergence under this condition by bounding the gradients along the optimization trajectory.
- We prove the convergence of *constant-stepsize* GD/SGD/NAG in the convex and non-convex settings, and obtain the classical rates for all of them, as summarized in Table 1.

Besides the generalized smoothness condition and the new approach, our results are also novel in the following aspects.

- The convergence results of *constant-stepsize* methods challenge the folklore belief on the necessity of adaptive stepsize for generalized smooth functions.
- We obtain new convergence results for GD and NAG in the convex setting under the generalized smoothness condition.
- We relax the assumption of bounded noise to the weaker one of bounded variance of noise in the stochastic setting with the simple SGD method.

## 2 Related work

**Gradient-based optimizaiton.** The classical gradient-based optimization problems for the standard Lipschitz smooth functions have been well studied for both convex [Nemirovskij and Yudin, 1983,

---

[2]This statement is informal because we can only bound Hessian norms *along the trajectory*, rather than almost everywhere within a convex set as in the standard Lipschitz smoothness condition. For example, even if the Hessian norm is bounded at both $x_t$ and $x_{t+1}$, it does not directly mean the Hessian norm is also bounded over the line segment between them, which is required in classical analysis. A more formal statement will need Lemma 3.3 presented later in the paper.

Table 1: Summary of the results. $\epsilon$ denotes the sub-optimality gap of the function value in convex settings, and the gradient norm in non-convex settings. "$*$" denotes optimal rates.

| Method | Convexity | $\ell$-smoothness | Gradient complexity |
|---|---|---|---|
| GD | Strongly convex Convex | No requirement | $\mathcal{O}(\log(1/\epsilon))$ (Theorem 4.3) $\mathcal{O}(1/\epsilon)$ (Theorem 4.2 ) |
| | Non-convex | Sub-quadratic $\ell$ | $\mathcal{O}(1/\epsilon^2)^*$ (Theorem 5.2) |
| | | Quadratic $\ell$ | $\Omega(\text{exp. in cond \#})$ (Theorem 5.4 ) |
| NAG | Convex | Sub-quadratic $\ell$ | $\mathcal{O}(1/\sqrt{\epsilon})^*$ (Theorem 4.4 ) |
| SGD | Non-convex | Sub-quadratic $\ell$ | $\mathcal{O}(1/\epsilon^4)^*$ (Theorem 5.3) |

Nesterov, 2003, d'Aspremont et al., 2021] and non-convex functions. In the convex setting, the goal is to reach an $\epsilon$-sub-optimal point $x$ satisfying $f(x) - \inf_x f(x) \leq \epsilon$. It is well known that GD achieves the $\mathcal{O}(1/\epsilon)$ gradient complexity and NAG achieves the accelerated $\mathcal{O}(1/\sqrt{\epsilon})$ complexity which is optimal among all gradient-based methods. For strongly convex functions, GD and NAG achieve the $\mathcal{O}(\kappa \log(1/\epsilon))$ and $\mathcal{O}(\sqrt{\kappa} \log(1/\epsilon))$ complexity respectively, where $\kappa$ is the condition number and the latter is again optimal. In the non-convex setting, the goal is to find an $\epsilon$-stationary point $x$ satisfying $\|\nabla f(x)\| \leq \epsilon$, since finding a global minimum is NP-hard in general. It is well known that GD achieves the optimal $\mathcal{O}(1/\epsilon^2)$ complexity which matches the lower bound in [Carmon et al., 2017]. In the stochastic setting for unbiased stochastic gradient with bounded variance, SGD achieves the optimal $\mathcal{O}(1/\epsilon^4)$ complexity [Ghadimi and Lan, 2013], matching the lower bound in [Arjevani et al., 2019]. In this paper, we obtain the classical rates in terms of $\epsilon$ for all the above-mentioned methods and settings, under a far more general smoothness condition.

**Generalized smoothness.** The $(L_0, L_1)$-smoothness condition proposed by Zhang et al. [2019] was studied by many follow-up works. Under the same condition, [Zhang et al., 2020] considers momentum in the updates and improves the constant dependency of the convergence rate for SGD with clipping derived in [Zhang et al., 2019]. [Qian et al., 2021] studies gradient clipping in incremental gradient methods, [Zhao et al., 2021] studies stochastic normalized gradient descent, and [Crawshaw et al., 2022] studies a generalized SignSGD method, under the $(L_0, L_1)$-smoothess condition. [Reisizadeh et al., 2023] studies variance reduction for $(L_0, L_1)$-smooth functions. [Chen et al., 2023] proposes a new notion of $\alpha$-symmetric generalized smoothness, which is roughly as general as $(L_0, L_1)$-smoothness. [Wang et al., 2022] analyzes convergence of Adam and provides a lower bound which shows non-adaptive SGD may diverge. In the stochastic setting, the above-mentioned works either consider the strong assumption of bounded gradient noise or require a very large batch size that depends on $\epsilon$, which essentially reduces the analysis to the deterministic setting. [Faw et al., 2023] proposes an AdaGrad-type algorithm in order to relax the bounded noise assumption. Perhaps due to the lower bounds in [Zhang et al., 2019, Wang et al., 2022], all the above works study methods with an adaptive stepsize. In this and our concurrent work [Li et al., 2023], we further generalize the smoothness condition and analyze various methods under this condition through bounding the gradients along the trajectory.

## 3 Function class

In this section, we discuss the function class of interest where the objective function $f$ lies. We start with the following two standard assumptions in the literature of unconstrained optimization, which will be assumed throughout Sections 4 and 5 unless explicitly stated.

**Assumption 1.** The objective function $f$ is differentiable and *closed* within its *open* domain $\mathcal{X}$.

**Assumption 2.** The objective function $f$ is bounded from below, i.e., $f^* := \inf_{x \in \mathcal{X}} f(x) > -\infty$.

A function $f$ is said to be closed if its sub-level set $\{x \in \text{dom}(f) \mid f(x) \leq a\}$ is closed for each $a \in \mathbb{R}$. A continuous function $f$ with an open domain is closed if and only $f(x)$ tends to positive infinity when $x$ approaches the boundary of its domain [Boyd and Vandenberghe, 2004]. Assumption 1 is necessary for our analysis to ensure that the iterates of a method with a reasonably small stepsize stays within the domain $\mathcal{X}$. Note that for $\mathcal{X} = \mathbb{R}^d$ considered in most unconstrained

optimization papers, the assumption is trivially satisfied as all continuous functions over $\mathbb{R}^d$ are closed. We consider a more general domain which may not be the whole space because that is the case for some interesting examples in our function class of interest (see Section 3.1.3). However, it actually brings us some additional technical difficulties especially in the stochastic setting, as we need to make sure the iterates do not go outside of the domain.

## 3.1 Generalized smoothness

In this section, we formally define the generalized smoothness condition, and present its properties and examples.

### 3.1.1 Definitions

Definitions 1 and 2 below are two equivalent ways of stating the definition, where we use $\mathcal{B}(x, R)$ to denote the Euclidean ball with radius $R$ centered at $x$.

**Definition 1** ($\ell$-smoothness). A real-valued differentiable function $f : \mathcal{X} \to \mathbb{R}$ is $\ell$-smooth for some non-decreasing continuous function $\ell : [0, +\infty) \to (0, +\infty)$ if $\left\|\nabla^2 f(x)\right\| \leq \ell(\|\nabla f(x)\|)$ *almost everywhere* (with respect to the Lebesgue measure) in $\mathcal{X}$.

*Remark* 3.1. Definition 1 reduces to the classical $L$-smoothness when $\ell \equiv L$ is a constant function. It reduces to the $(L_0, L_1)$-smoothness proposed in [Zhang et al., 2019] when $\ell(u) = L_0 + L_1 u$ is an affine function.

**Definition 2** ($(r, \ell)$-smoothness). A real-valued differentiable function $f : \mathcal{X} \to \mathbb{R}$ is $(r, \ell)$-smooth for continuous functions $r, \ell : [0, +\infty) \to (0, +\infty)$ where $\ell$ is non-decreasing and $r$ is non-increasing, if it satisfies 1) for any $x \in \mathcal{X}$, $\mathcal{B}(x, r(\|\nabla f(x)\|)) \subseteq \mathcal{X}$, and 2) for any $x_1, x_2 \in \mathcal{B}(x, r(\|\nabla f(x)\|))$, $\|\nabla f(x_1) - \nabla f(x_2)\| \leq \ell(\|\nabla f(x)\|) \cdot \|x_1 - x_2\|$.

The requirements that $\ell$ is non-decreasing and $r$ is non-increasing do not cause much loss in generality. If these conditions are not satisfied, one can replace $\ell$ and $r$ with the non-increasing function $\tilde{r}(u) := \inf_{0 \leq v \leq u} r(v) \leq r(u)$ and non-decreasing function $\tilde{\ell}(u) := \sup_{0 \leq v \leq u} \ell(v) \geq \ell(u)$ in Definitions 1 and 2. Then the only requirement is $\tilde{r} > 0$ and $\tilde{\ell} < \infty$.

Next, we prove that the above two definitions are equivalent in the following proposition, whose proof is involved and deferred to Appendix A.2.

**Proposition 3.2.** *An $(r, \ell)$-smooth function is $\ell$-smooth; and an $\ell$-smooth function satisfying Assumption 1 is $(r, m)$-smooth where $m(u) := \ell(u + a)$ and $r(u) := a/m(u)$ for any $a > 0$.*

The condition in Definition 1 is simple and one can easily check whether it is satisfied for a given example function. On the other hand, Definition 2 is a local Lipschitz condition on the gradient that is harder to verify. However, it is useful for deriving several useful properties in the next section.

### 3.1.2 Properties

First, we provide the following lemma which is very useful in our analyses of all the methods considered in this paper. Its proof is deferred to Appendix A.3.

**Lemma 3.3.** *If $f$ is $(r, \ell)$-smooth, for any $x \in \mathcal{X}$ satisfying $\|\nabla f(x)\| \leq G$, we have 1) $\mathcal{B}(x, r(G)) \subseteq \mathcal{X}$, and 2) for any $x_1, x_2 \in \mathcal{B}(x, r(G))$,*

$$\|\nabla f(x_1) - \nabla f(x_2)\| \leq L \|x_1 - x_2\|, \quad f(x_1) \leq f(x_2) + \langle \nabla f(x_2), x_1 - x_2 \rangle + \frac{L}{2} \|x_1 - x_2\|^2, \quad (2)$$

*where $L := \ell(G)$ is the effective smoothness constant.*

*Remark* 3.4. Since we have shown the equivalence between $\ell$-smoothness and $(r, \ell)$-smoothness, Lemma 3.3 also applies to $\ell$-smooth functions, for which we have $L = \ell(2G)$ and $r(G) = G/L$ if choosing $a = G$ in Proposition 3.2.

Lemma 3.3 states that, if the gradient at $x$ is bounded by some constant $G$, then within its neighborhood with a *constant* radius, we can obtain (2), the same inequalities that were derived in the textbook analysis [Nesterov, 2003] under the standard Lipschitz smoothness condition. With (2), the analysis for generalized smoothness is not much harder than that for standard smoothness. Since we

Table 2: Examples of univariate $(\rho, L_0, L_\rho)$ smooth functions for different $\rho$s. The parameters $a, b, p$ are *real numbers* (not necessarily integers) satisfying $a, b > 1$ and $p < 1$ or $p \geq 2$. We use $1^+$ to denote any real number slightly larger than 1.

| $\rho$ | 0 | 1 | 1 | $1^+$ | 1.5 | 2 | $\frac{p-2}{p-1}$ |
|---|---|---|---|---|---|---|---|
| Example Functions | Quadratic | Polynomial | $a^x$ | $a^{(b^x)}$ | Rational | Logarithmic | $x^p$ |

mostly choose $x = x_2 = x_t$ and $x_1 = x_{t+1}$ in the analysis, in order to apply Lemma 3.3, we need two conditions: $\|\nabla f(x_t)\| \leq G$ and $\|x_{t+1} - x_t\| \leq r(G)$ for some constant $G$. The latter is usually directly implied by the former for most deterministic methods with a small enough stepsize, and the former can be obtained with our new approach that bounds the gradients along the trajectory.

With Lemma 3.3, we can derive the following useful lemma which is the reverse direction of a generalized Polyak-Lojasiewicz (PL) inequality, whose proof is deferred to Appendix A.3.

**Lemma 3.5.** *If $f$ is $\ell$-smooth, then $\|\nabla f(x)\|^2 \leq 2\ell(2\|\nabla f(x)\|) \cdot (f(x) - f^*)$ for any $x \in \mathcal{X}$.*

Lemma 3.5 provides an inequality involving the gradient norm and the sub-optimality gap. For example, when $\ell(u) = u^\rho$ for some $0 \leq \rho < 2$, this lemma suggests $\|\nabla f(x)\| \leq \mathcal{O}\left((f(x) - f^*)^{1/(2-\rho)}\right)$, which means the gradient norm is bounded whenever the function value is bounded. The following corollary provides a more formal statement for general sub-quadratic $\ell$ (i.e., $\lim_{u \to \infty} \ell(u)/u^2 = 0$), and we defer its proof to Appendix A.3.

**Corollary 3.6.** *Suppose $f$ is $\ell$-smooth where $\ell$ is sub-quadratic. If $f(x) - f^* \leq F$ for some $x \in \mathcal{X}$ and $F \geq 0$, denoting $G := \sup\{u \geq 0 \mid u^2 \leq 2\ell(2u) \cdot F\}$, then they satisfy $G^2 = 2\ell(2G) \cdot F$ and we have $\|\nabla f(x)\| \leq G < \infty$.*

Therefore, in order to bound the gradients along the trajectory as we discussed below Lemma 3.3, it suffices to bound the function values, which is usually easier.

### 3.1.3 Examples

The most important subset of $\ell$-smooth (or $(r, \ell)$-smooth) functions are those with a polynomial $\ell$, and can be characterized by the $(\rho, L_0, L_\rho)$-smooth function class defined below.

**Definition 3** ($(\rho, L_0, L_\rho)$-smoothness). A real-valued differentiable function $f$ is $(\rho, L_0, L_\rho)$-smooth for constants $\rho, L_0, L_\rho \geq 0$ if it is $\ell$-smooth with $\ell(u) = L_0 + L_\rho u^\rho$.

Definition 3 reduces to the standard Lipschitz smoothness condition when $\rho = 0$ or $L_\rho = 0$ and to the $(L_0, L_1)$-smoothness proposed in [Zhang et al., 2019] when $\rho = 1$. We list several univariate examples of $(\rho, L_0, L_\rho)$-smooth functions for different $\rho$s in Table 2 with their rigorous justifications in Appendix A.1. Note that when $x$ goes to infinity, polynomial and exponential functions corresponding to $\rho = 1$ grow much faster than quadratic functions corresponding to $\rho = 0$. Rational and logarithmic functions for $\rho > 1$ grow even faster as they can blow up to infinity near finite points. Note that the domains of such functions are not $\mathbb{R}^d$, which is why we consider the more general Assumption 1 instead of simply assuming $\mathcal{X} = \mathbb{R}^d$.

Aside from logarithmic functions, the $(2, L_0, L_2)$-smooth function class also includes other univariate *self-concordant* functions. This is an important function class in the analysis of Interior Point Methods and coordinate-free analysis of the Newton method [Nesterov, 2003]. More specifically, a convex function $h : \mathbb{R} \to \mathbb{R}$ is self-concordant if $|h'''(x)| \leq 2h''(x)^{3/2}$ for all $x \in \mathbb{R}$. Formally, we have the following proposition whose proof is deferred to Appendix A.1.

**Proposition 3.7.** *If $h : \mathbb{R} \to \mathbb{R}$ is a self-concordant function satisfying $h''(x) > 0$ over the interval $(a, b)$, then $h$ restricted on $(a, b)$ is $(2, L_0, 2)$-smooth for some $L_0 > 0$.*

## 4 Convex setting

In this section, we present the convergence results of gradient descent (GD) and Nesterov's accelerated gradient method (NAG) in the convex setting. Formally, we define convexity as follows.

**Definition 4.** A real-valued differentiable function $f : \mathcal{X} \to \mathbb{R}$ is $\mu$-strongly-convex for $\mu \geq 0$ if $\mathcal{X}$ is a convex set and $f(y) - f(x) \geq \langle \nabla f(x), y - x \rangle + \frac{\mu}{2} \|y - x\|^2$ for any $x, y \in \mathcal{X}$. A function is convex if it is $\mu$-strongly-convex with $\mu = 0$.

We assume the existence of a global optimal point $x^*$ throughout this section, as in the following assumption. However, we want to note that, for gradient descent, this assumption is just for simplicity rather than necessary.

**Assumption 3.** There exists a point $x^* \in \mathcal{X}$ such that $f(x^*) = f^* = \inf_{x \in \mathcal{X}} f(x)$.

## 4.1 Gradient descent

The gradient descent method with a constant stepsize $\eta$ is defined via the following update rule

$$x_{t+1} = x_t - \eta \nabla f(x_t). \tag{3}$$

As discussed below Lemma 3.3, the key in the convergence analysis is to show $\|\nabla f(x_t)\| \leq G$ for all $t \geq 0$ and some constant $G$. We will prove it by induction relying on the following lemma whose proof is deferred to Appendix B.

**Lemma 4.1.** *For any $x \in \mathcal{X}$ satisfying $\|\nabla f(x)\| \leq G$, define $x^+ := x - \eta \nabla f(x)$. If $f$ is convex and $(r, \ell)$-smooth, and $\eta \leq \min\left\{ \frac{2}{\ell(G)}, \frac{r(G)}{2G} \right\}$, we have $x^+ \in \mathcal{X}$ and $\|\nabla f(x^+)\| \leq \|\nabla f(x)\| \leq G$.*

Lemma 4.1 suggests that for gradient descent (3) with a small enough stepsize, if the gradient norm at $x_t$ is bounded by $G$, then we have $\|\nabla f(x_{t+1})\| \leq \|\nabla f(x_t)\| \leq G$, i.e., the gradient norm is also bounded by $G$ at $t+1$. In other words, the gradient norm is indeed a non-increasing potential function for gradient descent in the convex setting. With a standard induction argument, we can show that $\|\nabla f(x_t)\| \leq \|\nabla f(x_0)\|$ for all $t \geq 0$. As discussed below Lemma 3.3, then we can basically apply the classical analysis to obtain the convergence guarantee in the convex setting as in the following theorem, whose proof is deferred to Appendix B.

**Theorem 4.2.** *Suppose $f$ is convex and $(r, \ell)$-smooth. Denote $G := \|\nabla f(x_0)\|$ and $L := \ell(G)$, then the iterates generated by (3) with $\eta \leq \min\left\{ \frac{1}{L}, \frac{r(G)}{2G} \right\}$ satisfy $\|\nabla f(x_t)\| \leq G$ for all $t \geq 0$ and*

$$f(x_T) - f^* \leq \frac{\|x_0 - x^*\|^2}{2\eta T}.$$

Since $\eta$ is a constant independent of $\epsilon$ or $T$, Theorem 4.2 achieves the classical $\mathcal{O}(1/T)$ rate, or $\mathcal{O}(1/\epsilon)$ gradient complexity to achieve an $\epsilon$-sub-optimal point, under the generalized smoothness condition. Since strongly convex functions are a subset of convex functions, Lemma 4.1 still holds for them. Then we immediately obtain the following result in the strongly convex setting, whose proof is deferred to Appendix B.

**Theorem 4.3.** *Suppose $f$ is $\mu$-strongly-convex and $(r, \ell)$-smooth. Denote $G := \|\nabla f(x_0)\|$ and $L := \ell(G)$, then the iterates generated by (3) with $\eta \leq \min\left\{ \frac{1}{L}, \frac{r(G)}{2G} \right\}$ satisfy $\|\nabla f(x_t)\| \leq G$ for all $t \geq 0$ and*

$$f(x_T) - f^* \leq \frac{\mu(1 - \eta\mu)^T}{2(1 - (1 - \eta\mu)^T)} \|x_0 - x^*\|^2.$$

Theorem 4.3 gives a linear convergence rate and the $\mathcal{O}((\eta\mu)^{-1} \log(1/\epsilon))$ gradient complexity to achieve an $\epsilon$-sub-optimal point. Note that for $\ell$-smooth functions, we have $\frac{r(G)}{G} = \frac{1}{L}$ (see Remark 3.4), which means we can choose $\eta = \frac{1}{2L}$. Then we obtain the $\mathcal{O}(\kappa \log(1/\epsilon))$ rate, where $\kappa := L/\mu$ is the local condition number around the initial point $x_0$. For standard Lipschitz smooth functions, it reduces to the classical rate of gradient descent.

## 4.2 Nesterov's accelerated gradient method

In the case of convex and standard Lipschitz smooth functions, it is well known that Nesterov's accelerated gradient method (NAG) achieves the optimal $\mathcal{O}(1/T^2)$ rate. In this section, we show that

---

| Algorithm 1: Nesterov's Accelerated Gradient Method (NAG) |
|---|

**input** A convex and $\ell$-smooth function $f$, stepsize $\eta$, initial point $x_0$
  1: **Initialize** $z_0 = x_0$, $B_0 = 0$, and $A_0 = 1/\eta$.
  2: **for** $t = 0, \ldots$ **do**
  3:    $B_{t+1} = B_t + \frac{1}{2}\left(1 + \sqrt{4B_t + 1}\right)$
  4:    $A_{t+1} = B_{t+1} + 1/\eta$
  5:    $y_t = x_t + (1 - A_t/A_{t+1})(z_t - x_t)$
  6:    $x_{t+1} = y_t - \eta \nabla f(y_t)$
  7:    $z_{t+1} = z_t - \eta(A_{t+1} - A_t)\nabla f(y_t)$
  8: **end for**

---

under the $\ell$-smoothness condition with a *sub-quadratic* $\ell$, the optimal $\mathcal{O}(1/T^2)$ rate can be achieved by a slightly modified version of NAG shown in Algorithm 1, the only difference between which and the classical NAG is that the latter directly sets $A_{t+1} = B_{t+1}$ in Line 4. Formally, we have the following theorem, whose proof is deferred to Appendix C.

**Theorem 4.4.** *Suppose $f$ is convex and $\ell$-smooth where $\ell$ is sub-quadratic. Then there always exists a constant $G$ satisfying $G \geq \max\left\{8\sqrt{\ell(2G)((f(x_0) - f^*) + \|x_0 - x^*\|^2)}, \|\nabla f(x_0)\|\right\}$. Denote $L := \ell(2G)$ and choose $\eta \leq \min\left\{\frac{1}{16L^2}, \frac{1}{2L}\right\}$. The iterates generated by Algorithm 1 satisfy*

$$f(x_T) - f^* \leq \frac{4(f(x_0) - f^*) + 4\|x_0 - x^*\|^2}{\eta T^2 + 4}.$$

It is easy to note that Theorem 4.4 achieves the accelerated $\mathcal{O}(1/T^2)$ convergence rate, or equivalently the $\mathcal{O}(1/\sqrt{\epsilon})$ gradient complexity to find an $\epsilon$-sub-optimal point, which is optimal among gradient-based methods [Nesterov, 2003].

In order to prove Theorem 4.4, we also use induction to show the gradients along the trajectory of Algorithm 1 are bounded by $G$. However, unlike gradient descent, the gradient norm is no longer a potential function or monotonically non-increasing, which makes the induction analysis more challenging. Suppose that we have shown $\|\nabla f(y_s)\| \leq G$ for $s < t$. To complete the induction, it suffices to prove $\|\nabla f(y_t)\| \leq G$. Since $x_t = y_{t-1} - \eta \nabla f(y_{t-1})$ is a gradient descent step by Line 6 of Algorithm 1, Lemma 4.1 directly shows $\|\nabla f(x_t)\| \leq G$. In order to also bound $\|\nabla f(y_t)\|$, we try to control $\|y_t - x_t\|$, which is the most challenging part of our proof. Since $y_t - x_t$ can be expressed as a linear combination of past gradients $\{\nabla f(y_s)\}_{s<t}$, it might grow linearly with $t$ if we simply apply $\|\nabla f(y_s)\| \leq G$ for $s < t$. Fortunately, Lemma 3.5 allows us to control the gradient norm with the function value. Thus if the function value is decreasing sufficiently fast, which can be shown by following the standard Lyapunov analysis of NAG, we are able to obtain a good enough bound on $\|\nabla f(y_s)\|$ for $s < t$, which allows us to control $\|y_t - x_t\|$. We defer the detailed proof to Appendix C.

Note that Theorem 4.4 requires a smaller stepsize $\eta = \mathcal{O}(1/L^2)$, compared to the classical $\mathcal{O}(1/L)$ stepsize for standard Lipschitz smooth functions. The reason is we require a small enough stepsize to get a good enough bound on $\|y_t - x_t\|$. However, if the function is further assumed to be $\ell$-smooth with a *sub-linear* $\ell$, the requirement of stepsize can be relaxed to $\eta = \mathcal{O}(1/L)$, similar to the classical requirement. See Appendix C for the details.

In the strongly convex setting, we can also prove convergence of NAG with different $\{A_t\}_{t \geq 0}$ parameters when $f$ is $\ell$-smooth with a sub-quadratic $\ell$, or $(\rho, L_0, L_\rho)$-smooth with $\rho < 2$. The rate can be further improved when $\rho$ becomes smaller. However, since the constants $G$ and $L$ are different for GD and NAG, it is not clear whether the rate of NAG is faster than that of GD in the strongly convex setting. We will present the detailed result and analysis in Appendix D.

## 5  Non-convex setting

In this section, we present convergence results of gradient descent (GD) and stochastic gradient descent (SGD) in the non-convex setting.

## 5.1 Gradient descent

Similar to the convex setting, we still want to bound the gradients along the trajectory. However, in the non-convex setting, the gradient norm is not necessarily non-increasing. Fortunately, similar to the classical analyses, the function value is still non-increasing and thus a potential function, as formally shown in the following lemma, whose proof is deferred to Appendix E.

**Lemma 5.1.** *Suppose $f$ is $\ell$-smooth where $\ell$ is sub-quadratic. For any given $F \geq 0$, let $G := \sup \left\{ u \geq 0 \mid u^2 \leq 2\ell(2u) \cdot F \right\}$ and $L := \ell(2G)$. For any $x \in \mathcal{X}$ satisfying $f(x) - f^* \leq F$, define $x^+ := x - \eta \nabla f(x)$ where $\eta \leq 2/L$, we have $x^+ \in \mathcal{X}$ and $f(x^+) \leq f(x)$.*

Then using a standard induction argument, we can show $f(x_t) \leq f(x_0)$ for all $t \geq 0$. According to Corollary 3.6, it implies bounded gradients along the trajectory. Therefore, we can show convergence of gradient descent as in the following theorem, whose proof is deferred to Appendix E.

**Theorem 5.2.** *Suppose $f$ is $\ell$-smooth where $\ell$ is sub-quadratic. Let $G := \sup \left\{ u \geq 0 \mid u^2 \leq 2\ell(2u) \cdot (f(x_0) - f^*) \right\}$ and $L := \ell(2G)$. If $\eta \leq 1/L$, the iterates generated by (3) satisfy $\|\nabla f(x_t)\| \leq G$ for all $t \geq 0$ and*

$$\frac{1}{T} \sum_{t<T} \|\nabla f(x_t)\|^2 \leq \frac{2(f(x_0) - f^*)}{\eta T}.$$

It is clear that Theorem 5.2 gives the classical $\mathcal{O}(1/\epsilon^2)$ gradient complexity to achieve an $\epsilon$-stationary point, which is optimal as it matches the lower bound in [Carmon et al., 2017].

## 5.2 Stochastic gradient descent

In this part, we present the convergence result for stochastic gradient descent defined as follows.

$$x_{t+1} = x_t - \eta g_t, \tag{4}$$

where $g_t$ is an estimate of the gradient $\nabla f(x_t)$. We consider the following standard assumption on the gradient noise $\epsilon_t := g_t - \nabla f(x_t)$.

**Assumption 4.** $\mathbb{E}_{t-1}[\epsilon_t] = 0$ and $\mathbb{E}_{t-1}\left[\|\epsilon_t\|^2\right] \leq \sigma^2$ for some $\sigma \geq 0$, where $\mathbb{E}_{t-1}$ denotes the expectation conditioned on $\{g_s\}_{s<t}$.

Under Assumption 4, we can obtain the following theorem.

**Theorem 5.3.** *Suppose $f$ is $\ell$-smooth where $\ell$ is sub-quadratic. For any $0 < \delta < 1$, we denote $F := 8(f(x_0) - f^* + \sigma)/\delta$ and $G := \sup\{u \geq 0 \mid u^2 \leq 2\ell(2u) \cdot F\} < \infty$. Denote $L := \ell(2G)$ and choose $\eta \leq \min\left\{\frac{1}{2L}, \frac{1}{4G\sqrt{T}}\right\}$ and $T \geq \frac{F}{\eta\epsilon^2}$ for any $\epsilon > 0$. Then with probability at least $1 - \delta$, the iterates generated by (4) satisfy $\|\nabla f(x_t)\| \leq G$ for all $t < T$ and*

$$\frac{1}{T} \sum_{t<T} \|\nabla f(x_t)\|^2 \leq \epsilon^2.$$

As we choose $\eta = \mathcal{O}(1/\sqrt{T})$, Theorem 5.3 gives the classical $\mathcal{O}(1/\epsilon^4)$ gradient complexity, where we ignore non-leading terms. This rate is optimal as it matches the lower bound in [Arjevani et al., 2019]. The key to its proof is again to bound the gradients along the trajectory. However, bounding gradients in the stochastic setting is much more challenging than in the deterministic setting, especially with the heavy-tailed noise in Assumption 4. We briefly discuss some of the challenges as well as our approach below and defer the detailed proof of Theorem 5.3 to Appendix F.

First, due to the existence of heavy-tailed gradient noise as considered in Assumption 4, neither the gradient nor the function values is non-increasing. The induction analyses we have used in the deterministic setting hardly work. In addition, to apply Lemma 3.3, we need to control the update at each step and make sure $\|x_{t+1} - x_t\| = \eta \|g_t\| \leq G/L$. However, $g_t$ might be unbounded due to the potentially unbounded gradient noise.

To overcome these challenges, we define the following random variable $\tau$.

$$\tau_1 := \min\{t \mid f(x_{t+1}) - f^* > F\} \wedge T,$$

$$\tau_2 := \min\left\{t \,\middle|\, \|\epsilon_t\| > \frac{G}{5\eta L}\right\} \wedge T, \tag{5}$$

$$\tau := \min\{\tau_1, \tau_2\},$$

where we use $a \wedge b$ to denote $\min\{a, b\}$ for any $a, b \in \mathbb{R}$. Then at least before time $\tau$, we know that the function value and gradient noise are bounded, where the former also implies bounded gradients according to Corollary 3.6. Therefore, it suffices to show the probability of $\tau < T$ is small, which means with a high probability, $\tau = T$ and thus gradients are always bounded before $T$.

Since both the gradient and noise are bounded for $t < \tau$, it is straightforward to bound the update $\|x_{t+1} - x_t\|$, which allows us to use Lemma 3.3 and other useful properties. However, it is still non-trivial to upper bound $\mathbb{E}[f(x_\tau) - f^*]$ as $\tau$ is a random variable instead of a fixed time step. Fortunately, $\tau$ is a stopping time with nice properties. That is because both $f(x_{t+1})$ and $\epsilon_t = g_t - \nabla f(x_t)$ only depend on $\{g_s\}_{s \leq t}$, i.e., the stochastic gradients up to $t$. Therefore, for any fixed $t$, the events $\{\tau > t\}$ only depend on $\{g_s\}_{s \leq t}$, which show $\tau$ is a stopping time. Then with a careful analysis, we are still able to obtain an upper bound on $\mathbb{E}[f(x_\tau) - f^*] = \mathcal{O}(1)$.

On the other hand, $\tau < T$ means either $\tau = \tau_1 < T$ or $\tau = \tau_2 < T$. If $\tau = \tau_1 < T$, by its definition, we know $f(x_{\tau+1}) - f^* > F$. Roughly speaking, it also suggests $f(x_\tau) - f^* > F/2$. If we choose $F$ such that it is much larger than the upper bound on $\mathbb{E}[f(x_\tau) - f^*]$ we just obtained, by Markov's inequality, we can show the probability of $\tau = \tau_1 < T$ is small. In addition, by union bound and Chebyshev's inequality, the probability of $\tau_2 < T$ can also be bounded by a small constant. Therefore, we have shown $\tau < T$. Then the rest of the analysis is not too hard following the classical analysis.

## 5.3 Reconciliation with existing lower bounds

In this section, we reconcile our convergence results for constant-stepsize GD/SGD in the non-convex setting with existing lower bounds in [Zhang et al., 2019] and [Wang et al., 2022], based on which the authors claim that adaptive methods such as GD/SGD with clipping and Adam are provably faster than non-adaptive GD/SGD. This may seem to contradict our convergence results. In fact, we show that any gain in adaptive methods is at most by constant factors, as GD and SGD already achieve the optimal rates in the non-convex setting.

[Zhang et al., 2019] provides both upper and lower complexity bounds for constant-stepsize GD for $(L_0, L_1)$-smooth functions, and shows that its complexity is $\mathcal{O}(M\epsilon^{-2})$, where

$$M := \sup\{\|\nabla f(x)\| \mid f(x) \leq f(x_0)\}$$

is the supremum gradient norm below the level set of the initial function value. If $M$ is very large, then the $\mathcal{O}(M\epsilon^{-2})$ complexity can be viewed as a negative result, and as evidence that constant-stepsize GD can be slower than GD with gradient clipping, since in the latter case, they obtain the $\mathcal{O}(\epsilon^{-2})$ complexity without $M$. However, based on our Corollary 3.6, their $M$ can be actually bounded by our $G$, which is a constant. Therefore, the gain in adaptive methods is at most by constant factors.

[Wang et al., 2022] further provides a lower bound which shows non-adaptive GD may diverge for some examples. However, their counter-example does not allow the stepsize to depend on the initial sub-optimality gap. In contrast, our stepsize $\eta$ depends on the effective smoothness constant $L$, which depends on the initial sub-optimality gap through $G$. Therefore, there is no contradiction here either. We should point out that in the practice of training neural networks, the stepsize is usually tuned after fixing the loss function and initialization, so it does depend on the problem instance and initialization.

## 5.4 Lower bound

For $(\rho, L_0, L_\rho)$-smooth functions with $\rho < 2$, it is easy to verify that the constant $G$ in both Theorem 5.2 and Theorem 5.3 is a polynomial function of problem-dependent parameters like $L_0, L_\rho, f(x_0) - f^*, \sigma$, etc. In other words, GD and SGD are provably efficient methods in the non-convex setting for $\rho < 2$. In this section, we show that the requirement of $\rho < 2$ is necessary in the non-convex setting with the lower bound for GD in the following Theorem 5.4, whose proof is deferred in Appendix G. Since SGD reduces to GD when there is no gradient noise, it is also a lower bound for SGD.

**Theorem 5.4.** *Given $L_0, L_2, G_0, \Delta_0 > 0$ satisfying $L_2\Delta_0 \geq 10$, for any $\eta \geq 0$, there exists a $(2, L_0, L_2)$-smooth function $f$ that satisfies Assumptions 1 and 2, and initial point $x_0$ that satisfies $\|\nabla f(x_0)\| \leq G_0$ and $f(x_0) - f^* \leq \Delta_0$, such that gradient descent with stepsize $\eta$ (3) either cannot reach a 1-stationary point or takes at least $\exp(L_2\Delta_0/8)/6$ steps to reach a 1-stationary point.*

## 6 Conclusion

In this paper, we generalize the standard Lipschitz smoothness as well as the $(L_0, L_1)$-smoothness [Zhang et al., 2020] conditions to the $\ell$-smoothness condition, and develop a new approach for analyzing the convergence under this condition. The approach uses different techniques for several methods and settings to bound the gradient along the optimization trajectory, which allows us to obtain stronger results for both convex and non-convex problems. We obtain the classical rates for GD/SGD/NAG methods in the convex and/or non-convex setting. Our results challenge the folklore belief on the necessity of adaptive methods for generalized smooth functions.

There are several interesting future directions following this work. First, the $\ell$-smoothness can perhaps be further generalized by allowing $\ell$ to also depend on potential functions in each setting, besides the gradient norm. In addition, it would also be interesting to see if the techniques of bounding gradients along the trajectory that we have developed in this and the concurrent work [Li et al., 2023] can be further generalized to other methods and problems and to see whether more efficient algorithms can be obtained. Finally, although we justified the necessity of the requirement of $\ell$-smoothness with a *sub-quadratic* $\ell$ in the non-convex setting, it is not clear whether it is also necessary for NAG in the convex setting, another interesting open problem.

## Acknowledgments

This work was supported, in part, by the MIT-IBM Watson AI Lab and ONR Grants N00014-20-1-2394 and N00014-23-1-2299. We also acknowledge support from DOE under grant DE-SC0022199, and NSF through awards DMS-2031883, DMS-1953181, and DMS-2022448 (TRIPODS program).

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

# A Proofs related to generalized smoothness

In this section, we provide the proofs of propositions and lemmas related to the generalized smoothness condition in Definition 1 or 2. First, in Appendix A.1, we justify the examples we discussed in Section 3. Next, we provide the detailed proof of Proposition 3.2 in Appendix A.2. Finally, we provide the proofs of the useful properties of generalized smoothness in Appendix A.3, including Lemma 3.3, Lemma 3.5, and Corollary 3.6 stated in Section 3.1.2.

## A.1 Justification of examples in Section 3

In this section, we justify the univariate examples of $(\rho, L_0, L_\rho)$-smooth functions listed in Table 2 and also provide the proof of Propositions 3.7.

First, it is well-known that all quadratic functions have bounded Hessian and are Lipschitz smooth, corresponding to $\rho = 0$. Next, [Zhang et al., 2019, Lemma 2] shows that any univariate polynomial is $(L_0, L_1)$-smooth, corresponding to $\rho = 1$. Then, regarding the exponential function $f(x) = a^x$ where $a > 1$, we have $f'(x) = \log(a)a^x$ and $f''(x) = \log(a)^2 a^x = \log(a)f'(x)$, which implies $f$ is $(1, 0, \log(a))$-smooth. Similarly, by standard calculations, it is straight forward to verify that logarithmic functions and $x^p, p \neq 1$ are also $(\rho, L_0, L_\rho)$-smooth with $\rho = 2$ and $\rho = \frac{p-2}{p-1}$ respectively.

So far we have justified all the examples in Table 2 except double exponential functions $a^{(b^x)}$ and rational functions, which will be justified rigorously by the two propositions below.

First, for double exponential functions in the form of $f(x) = a^{(b^x)}$ where $a, b > 1$, we have the following proposition, which shows $f$ is $(\rho, L_0, L_\rho)$-smooth for any $\rho > 1$.

**Proposition A.1.** *For any $\rho > 1$, the double exponential function $f(x) = a^{(b^x)}$, where $a, b > 1$, is $(\rho, L_0, L_\rho)$-smooth for some $L_0, L_\rho \geq 0$. However, it is not necessarily $(L_0, L_1)$-smooth for any $L_0, L_1 \geq 0$.*

*Proof of Proposition A.1.* By standard calculations, we can obtain

$$f'(x) = \log(a)\log(b)\, b^x a^{(b^x)}, \quad f''(x) = \log(b)(\log(a)b^x + 1) \cdot f'(x). \tag{6}$$

Note that if $\rho > 1$,

$$\lim_{x \to +\infty} \frac{|f'(x)|^\rho}{|f''(x)|} = \lim_{x \to +\infty} \frac{|f'(x)|^{\rho-1}}{\log(b)(\log(a)b^x + 1)} = \lim_{y \to +\infty} \frac{(\log(a)\log(b)y)^{\rho-1}\, a^{(\rho-1)y}}{\log(b)(\log(a)y + 1)} = \infty,$$

where the first equality is a direct calculation based on (6); the second equality uses change of variable $y = b^x$; and the last equality is because exponential functions grow faster than affine functions. Therefore, for any $L_\rho > 0$, there exists $x_0 \in \mathbb{R}$ such that $|f''(x)| \leq L_\rho |f'(x)|^\rho$ if $x > x_0$. Next, note that $\lim_{x \to -\infty} f''(x) = 0$. Then for any $\lambda_1 > 0$, there exists $x_1 \in \mathbb{R}$ such that $|f''(x)| \leq \lambda_1$ if $x < x_1$. Also, since $f''$ is continuous, by Weierstrass's Theorem, we have $|f''(x)| \leq \lambda_2$ if $x_1 \leq x \leq x_0$ for some $\lambda_2 > 0$. Then denoting $L_0 = \max\{\lambda_1, \lambda_2\}$, we know $f$ is $(\rho, L_0, L_\rho)$-smooth.

Next, to show $f$ is not necessarily $(L_0, L_1)$-smooth, consider the specific double exponential function $f(x) = e^{(e^x)}$. Then we have

$$f'(x) = e^x e^{(e^x)}, \quad f''(x) = (e^x + 1) \cdot f'(x).$$

For any $x \geq \max\{\log(L_0 + 1), \log(L_1 + 1)\}$, we can show that

$$|f''(x)| > (L_1 + 1)f'(x) > L_0 + L_1 |f'(x)|,$$

which shows $f$ is not $(L_0, L_1)$ smooth for any $L_0, L_1 \geq 0$. $\square$

In the next proposition, we show that any univariate rational function $f(x) = P(x)/Q(x)$, where $P$ and $Q$ are two polynomials, is $(\rho, L_0, L_\rho)$-smooth with $\rho = 1.5$.

**Proposition A.2.** *The rational function $f(x) = P(x)/Q(x)$, where $P$ and $Q$ are two polynomials, is $(1.5, L_0, L_{1.5})$-smooth for some $L_0, L_{1.5} \geq 0$. However, it is not necessarily $(\rho, L_0, L_\rho)$-smooth for any $\rho < 1.5$ and $L_0, L_\rho \geq 0$.*

*Proof of Proposition A.2.* Let $f(x) = P(x)/Q(x)$ where $P$ and $Q$ are two polynomials. Then the partial fractional decomposition of $f(x)$ is given by

$$f(x) = w(x) + \sum_{i=1}^{m} \sum_{r=1}^{j_i} \frac{A_{ir}}{(x - a_i)^r} + \sum_{i=1}^{n} \sum_{r=1}^{k_i} \frac{B_{ir}x + C_{ir}}{(x^2 + b_i x + c_i)^r},$$

where $w(x)$ is a polynomial, $A_{ir}, B_{ir}, C_{ir}, a_i, b_i, c_i$ are all real constants satisfying $b_i^2 - 4c_i < 0$ for each $1 \le i \le n$ which implies $x^2 + b_i x + c_i > 0$ for all $x \in \mathbb{R}$. Assume $j_i \ge 1$ and $A_{ij_i} \ne 0$ without loss of generality. Then we know $f$ has only finite singular points $\{a_i\}_{1 \le i \le m}$ and has continuous first and second order derivatives at all other points. To simplify notation, denote

$$p_{ir}(x) := \frac{A_{ir}}{(x - a_i)^r}, \quad q_{ir}(x) := \frac{B_{ir}x + C_{ir}}{(x^2 + b_i x + c_i)^r}.$$

Then we have $f(x) = w(x) + \sum_{i=1}^{m} \sum_{r=1}^{j_i} p_{ir}(x) + \sum_{i=1}^{n} \sum_{r=1}^{k_i} q_{ir}(x)$. We know that $\frac{r+2}{r+1} \le 1.5$ for any $r \ge 1$. Then we can show that

$$\lim_{x \to a_i} \frac{|f'(x)|^{1.5}}{|f''(x)|} = \lim_{x \to a_i} \frac{\left|p'_{ij_i}(x)\right|^{1.5}}{\left|p''_{ij_i}(x)\right|} \ge \frac{1}{j_i + 1}, \tag{7}$$

where the first equality is because one can easily verify that the first and second order derivatives of $p_{ij_i}$ dominate those of all other terms when $x$ goes to $a_i$, and the second equality is by standard calculations noting that $\frac{j_i+2}{j_i+1} \le 1.5$. Note that (7) implies that, for any $L_\rho > j_i + 1$, there exists $\delta_i > 0$ such that

$$|f''(x)| \le L_\rho |f'(x)|^{1.5}, \quad \text{if } |x - a_i| < \delta_i. \tag{8}$$

Similarly, one can show $\lim_{x \to \infty} \frac{|f'(x)|^{1.5}}{|f''(x)|} = \infty$, which implies there exists $x_0 > 0$ such that

$$|f''(x)| \le L_\rho |f'(x)|^{1.5}, \quad \text{if } |x| > x_0. \tag{9}$$

Define

$$\mathcal{B} := \{x \in \mathbb{R} \mid |x| \le x_0 \text{ and } |x - a_i| \ge \delta_i, \forall i\}.$$

We know $\mathcal{B}$ is a compact set and therefore the continuous function $f''$ is bounded within $\mathcal{B}$, i.e., there exists some constant $L_0 > 0$ such that

$$|f''(x)| \le L_0, \quad \text{if } x \in \mathcal{B}. \tag{10}$$

Combining (8), (9), and (10), we have shown

$$|f''(x)| \le L_0 + L_\rho |f'(x)|^{1.5}, \quad \forall x \in \text{dom}(f),$$

which completes the proof of the first part.

For the second part, consider the ration function $f(x) = 1/x$. Then we know that $f'(x) = -1/x^2$ and $f''(x) = 2/x^3$. Note that for any $\rho < 1.5$ and $0 < x \le \min\{(L_0 + 1)^{-1/3}, (L_\rho + 1)^{-1/(3-2\rho)}\}$, we have

$$|f''(x)| = \frac{1}{x^3} + \frac{1}{x^{3-2\rho}} \cdot |f'(x)|^\rho > L_0 + L_\rho |f'(x)|^\rho,$$

which shows $f$ is not $(\rho, L_0, L_\rho)$ smooth for any $\rho < 1.5$ and $L_0, L_\rho \ge 0$. $\qquad\square$

Finally, we complete this section with the proof of Proposition 3.7, which shows self-concordant functions are $(2, L_0, L_2)$-smooth for some $L_0, L_\rho \ge 0$.

*Proof of Proposition 3.7.* Let $h : \mathbb{R} \to \mathbb{R}$ be a self-concordant function. We have $h'''(x) \le 2h''(x)^{3/2}$. Then, for $x \in (a, b)$, we can obtain

$$\frac{1}{2} h''(x)^{-1/2} h'''(x) \le h''(x).$$

Integrating both sides from $x_0$ to $y$ for $x_0, y \in (a, b)$, we have

$$h''(y)^{1/2} - h''(x_0)^{1/2} \le h'(y) - h'(x_0).$$

Therefore,

$$h''(y) \le (h''(x_0)^{1/2} - h'(x_0) + h'(y))^2 \le 2(h''(x_0)^{1/2} - h'(x_0))^2 + 2h'(y)^2.$$

Since $h''(y) > 0$, we have $|h''(y)| = h''(y)$. Therefore, the above inequality shows that $h$ is $(2, L_0, L_2)$-smooth with $L_0 = 2(h''(x_0)^{1/2} - h'(x_0))^2$ and $L_2 = 2$. $\qquad\square$

## A.2 Proof of Proposition 3.2

In order to prove Proposition 3.2, we need the following several lemmas. First, the lemma below partially generalizes Grönwall's inequality.

**Lemma A.3.** *Let $\alpha : [a, b] \to [0, \infty)$ and $\beta : [0, \infty) \to (0, \infty)$ be two continuous functions. Suppose $\alpha'(t) \leq \beta(\alpha(t))$ almost everywhere over $(a, b)$. Denote function $\phi(u) := \int \frac{1}{\beta(u)} du$. We have for all $t \in [a, b]$,*

$$\phi(\alpha(t)) \leq \phi(\alpha(a)) - a + t.$$

*Proof of Lemma A.3.* First, by definition, we know that $\phi$ is increasing since $\phi' = \frac{1}{\beta} > 0$. Let function $\gamma : [a, b] \to \mathbb{R}$ be the solution of the following differential equation

$$\gamma'(t) = \beta(\gamma(t)) \ \forall t \in (a, b), \quad \gamma(a) = \alpha(a). \tag{11}$$

Then we have

$$d\phi(\gamma(t)) = \frac{d\gamma(t)}{\beta(\gamma(t))} = dt.$$

Integrating both sides, noting that $\gamma(a) = \alpha(a)$ by (11), we obtain

$$\phi(\gamma(t)) - \phi(\alpha(a)) = t - a.$$

Then it suffices to show $\phi(\alpha(t)) \leq \phi(\gamma(t)), \ \forall t \in [a, b]$. Note that the following inequality holds almost everywhere.

$$(\phi(\alpha(t)) - \phi(\gamma(t)))' = \phi'(\alpha(t))\alpha'(t) - \phi'(\gamma(t))\gamma'(t) = \frac{\alpha'(t)}{\beta(\alpha(t))} - \frac{\gamma'(t)}{\beta(\gamma(t))} \leq 0,$$

where the inequality is because $\alpha'(t) \leq \beta(\alpha(t))$ by the assumption of this lemma and $\gamma'(t) = \beta(\gamma(t))$ by (11). Since $\phi(\alpha(a)) - \phi(\gamma(a)) = 0$, we know for all $t \in [a, b]$, $\phi(\alpha(t)) \leq \phi(\gamma(t))$, which completes the proof. $\qquad\square$

With Lemma A.3, one can bound the gradient norm within a small enough neighborhood of a given point as in the following lemma.

**Lemma A.4.** *If the objective function $f$ is $\ell$-smooth, for any two points $x, y \in \mathbb{R}^d$ such that the closed line segment between $x$ and $y$ is contained in $\mathcal{X}$, if $\|y - x\| \leq \frac{a}{\ell(\|\nabla f(x)\| + a)}$ for any $a > 0$, we have*

$$\|\nabla f(y)\| \leq \|\nabla f(x)\| + a.$$

*Proof of Lemma A.4.* Denote $z(t) := (1 - t)x + ty$ for $0 \leq t \leq 1$. Then we know $z(t) \in \mathcal{X}$ for all $0 \leq t \leq 1$ by the assumption made in this lemma. Then we can also define $\alpha(t) := \|\nabla f(z(t))\|$ for $0 \leq t \leq 1$. Note that for any $0 \leq t \leq s \leq 1$, by triangle inequality,

$$\alpha(s) - \alpha(t) \leq \|\nabla f(z(s)) - \nabla f(z(t))\|. \tag{12}$$

We know that $\alpha(t) = \|\nabla f(z(t))\|$ is differentiable almost everywhere since $f$ is second order differentiable almost everywhere (Here we assume $\alpha(t) \neq 0$ for $0 < t < 1$ without loss of generality. Otherwise, one can define $t_m = \sup\{0 < t < 1 \mid \alpha(t) = 0\}$ and consider the interval $[t_m, 1]$ instead). Then the following equality holds almost everywhere

$$\alpha'(t) = \lim_{s \downarrow t} \frac{\alpha(s) - \alpha(t)}{s - t} \leq \lim_{s \downarrow t} \frac{\|\nabla f(z(s)) - \nabla f(z(t))\|}{s - t} = \left\|\lim_{s \downarrow t} \frac{\nabla f(z(s)) - \nabla f(z(t))}{s - t}\right\|$$

$$= \left\|\nabla^2 f(z(t))(y - x)\right\| \leq \left\|\nabla^2 f(z(t))\right\| \|y - x\| \leq \ell(\alpha(t)) \|y - x\|,$$

where the first inequality is due to (12) and the last inequality is by Definition 1. Let $\beta(u) := \ell(u) \cdot \|y - x\|$ and $\phi(u) := \int_0^u \frac{1}{\beta(v)} dv$. By Lemma A.3, we know that

$$\phi(\|\nabla f(y)\|) = \phi(u(1)) \leq \phi(u(0)) + 1 = \phi(\|\nabla f(x)\|) + 1.$$

Denote $\psi(u) := \int_0^u \frac{1}{\ell(v)} dv = \phi(u) \cdot \|y - x\|$. We have

$$
\begin{aligned}
\psi\left(\|\nabla f(y)\|\right) &\leq \psi\left(\|\nabla f(x)\|\right) + \|y - x\| \\
&\leq \psi\left(\|\nabla f(x)\|\right) + \frac{a}{\ell(\|\nabla f(x)\| + a)} \\
&\leq \int_0^{\|\nabla f(x)\|} \frac{1}{\ell(v)} dv + \int_{\|\nabla f(x)\|}^{\|\nabla f(x)\| + a} \frac{1}{\ell(v)} dv \\
&= \psi(\|\nabla f(x)\| + a).
\end{aligned}
$$

Since $\psi$ is increasing, we have $\|\nabla f(y)\| \leq \|\nabla f(x)\| + a$. $\qquad \square$

With Lemma A.4, we are ready to prove Proposition 3.2.

*Proof of Proposition 3.2.* We prove the two directions in this proposition separately.

**1. An $(r, \ell)$-smooth function is $\ell$-smooth.**

For each fixed $x \in \mathcal{X}$ where $\nabla^2 f(x)$ exists and any unit-norm vector $w$, by Definition 2, we know that for any $t \leq r(\|\nabla f(x)\|)$,

$$
\|\nabla f(x + tw) - \nabla f(x)\| \leq t \cdot \ell(\|\nabla f(x)\|).
$$

Then we know that

$$
\begin{aligned}
\left\|\nabla^2 f(x) w\right\| &= \left\|\lim_{t \downarrow 0} \frac{1}{t} (\nabla f(x + tw) - \nabla f(x))\right\| \\
&= \lim_{t \downarrow 0} \frac{1}{t} \|(\nabla f(x + tw) - \nabla f(x))\| \leq \ell(\|\nabla f(x)\|),
\end{aligned}
$$

which implies $\left\|\nabla^2 f(x)\right\| \leq \ell(\|\nabla f(x)\|)$ for any point $x$ if $\nabla^2 f(x)$ exists.

Then it suffices to show that $\nabla^2 f(x)$ exists almost everywhere. Note that for each $x \in \mathcal{X}$, Definition 2 states that the gradient function is $\ell(\|\nabla f(x)\|)$ Lipschitz within the ball $\mathcal{B}(x, r(\|\nabla f(x)\|))$. Then by Rademacher's Theorem, $f$ is twice differentiable almost everywhere within this ball. Then we can show it is also twice differentiable almost everywhere within the entire domain $\mathcal{X}$ as long as we can cover $\mathcal{X}$ with countably many such balls. Define $\mathcal{S}_n := \{x \in \mathcal{X} \mid n \leq \|\nabla f(x)\| \leq n + 1\}$ for integer $n \geq 0$. We have $\mathcal{X} = \cup_{n \geq 0} \mathcal{S}_n$. One can easily find an internal covering of $\mathcal{S}_n$ with balls of size $r(n+1)^3$, i.e., there exist $\{x_{n,i}\}_{i \geq 0}$, where $x_{n,i} \in \mathcal{S}_n$, such that $\mathcal{S}_n \subseteq \cup_{i \geq 0} \mathcal{B}(x_{n,i}, r(n+1)) \subseteq \cup_{i \geq 0} \mathcal{B}(x_{n,i}, r(\|\nabla f(x_{n,i})\|))$. Therefore we have $\mathcal{X} \subseteq \cup_{n,i \geq 0} \mathcal{B}(x_{n,i}, r(\|\nabla f(x_{n,i})\|))$ which completes the proof.

**2. An $\ell$-smooth function satisfying Assumption 1 is $(r, m)$-smooth where $m(u) := \ell(u + a)$ and $r(u) := a/m(u)$ for any $a > 0$.**

For any $y \in \mathbb{R}^d$ satisfying $\|y - x\| \leq r(\|\nabla f(x)\|) = \frac{a}{\ell(\|\nabla f(x)\| + a)}$, denote $z(t) := (1 - t)x + ty$ for $0 \leq t \leq 1$. We first show $y \in \mathcal{X}$ by contradiction. Suppose $y \notin \mathcal{X}$, let us define $t_{\mathsf{b}} := \inf\{0 \leq t \leq 1 \mid z(t) \notin \mathcal{X}\}$ and $z_{\mathsf{b}} := z(t_{\mathsf{b}})$. Then we know $z_{\mathsf{b}}$ is a boundary point of $\mathcal{X}$. Since $f$ is a closed function with an open domain, we have

$$
\lim_{t \uparrow t_{\mathsf{b}}} f(z(t)) = \infty. \tag{13}
$$

On the other hand, by the definition of $t_{\mathsf{b}}$, we know $z(t) \in \mathcal{X}$ for every $0 \leq t < t_{\mathsf{b}}$. Then by Lemma A.4, for all $0 \leq t < t_{\mathsf{b}}$, we have $\|\nabla f(z(t))\| \leq \|\nabla f(x)\| + a$. Therefore for all $0 \leq t < t_{\mathsf{b}}$,

$$
\begin{aligned}
f(z(t)) &\leq f(x) + \int_0^t \langle \nabla f(z(s)), y - x \rangle \, ds \\
&\leq f(x) + (\|\nabla f(x)\| + a) \cdot \|y - x\| \\
&< \infty,
\end{aligned}
$$

---

[3]We can find an internal covering in the following way. We first cover $\mathcal{S}_n$ with countably many hyper-cubes of length $r(n+1)/\sqrt{d}$, which is obviously doable. Then for each hyper-cube that intersects with $\mathcal{S}_n$, we pick one point from the intersection. Then the ball centered at the picked point with radius $r(n+1)$ covers this hyper-cube. Therefore, the union of all such balls can cover $\mathcal{S}_n$.

which contradicts (13). Therefore we have shown $y \in \mathcal{X}$. Since $y$ is chosen arbitrarily with the ball $\mathcal{B}(x, r(\|\nabla f(x)\|))$, we have $\mathcal{B}(x, r(\|\nabla f(x)\|)) \subseteq \mathcal{X}$. Then for any $x_1, x_2 \in \mathcal{B}(x, r(\|\nabla f(x)\|))$, we denote $w(t) := tx_1 + (1-t)x_2$. Then we know $w(t) \in \mathcal{B}(x, r(\|\nabla f(x)\|))$ for all $0 \leq t \leq 1$ and can obtain

$$
\begin{aligned}
\|\nabla f(x_1) - \nabla f(x_2)\| &= \left\| \int_0^1 \nabla^2 f(w(t)) \cdot (x_1 - x_2) \, dt \right\| \\
&\leq \|x_1 - x_2\| \cdot \int_0^1 \ell(\|\nabla f(x)\| + a) \, dt \\
&= m(\|\nabla f(x)\|) \cdot \|x_1 - x_2\|,
\end{aligned}
$$

where the last inequality is due to Lemma A.4. $\qquad \square$

### A.3 Proofs of lemmas implied by generalized smoothness

In this part, we provide the proofs of the useful properties stated in Section 3.1.2, including Lemma 3.3, Lemma 3.5, and Corollary 3.6.

*Proof of Lemma 3.3.* First, note that since $\ell$ is non-decreasing and $r$ is non-increasing, we have $\ell(\|\nabla f(x)\|) \leq \ell(G) = L$ and $r(G) \leq r(\|\nabla f(x)\|)$. Then by Definition 2, we directly have that $\mathcal{B}(x, r(G)) \subseteq \mathcal{B}(x, r(\|\nabla f(x)\|)) \subseteq \mathcal{X}$, and that for any $x_1, x_2 \in \mathcal{B}(x, r(G))$, we have

$$
\|\nabla f(x_1) - \nabla f(x_2)\| \leq \ell(\|\nabla f(x)\|) \|x_1 - x_2\| \leq L \|x_1 - x_2\|.
$$

Next, for the second inequality in (2), define $z(t) := (1-t)x_2 + tx_1$ for $0 \leq t \leq 1$. We know $z(t) \in \mathcal{B}(x, r(G))$. Note that we have shown

$$
\|\nabla f(z(t)) - \nabla f(x_2)\| \leq L \|z(t) - x_2\| = tL \|x_1 - x_2\|. \tag{14}
$$

Then we have

$$
\begin{aligned}
f(x_1) - f(x_2) &= \int_0^1 \langle \nabla f(z(t), x_1 - x_2 \rangle \, dt \\
&= \int_0^1 \langle \nabla f(x_2), x_1 - x_2 \rangle + \langle \nabla f(z(t)) - \nabla f(x_2), x_1 - x_2 \rangle \, dt \\
&\leq \langle \nabla f(x_2), x_1 - x_2 \rangle + L \|x_1 - x_2\|^2 \int_0^1 t \, dt \\
&= \langle \nabla f(x_2), x_1 - x_2 \rangle + \frac{L}{2} \|x_1 - x_2\|^2,
\end{aligned}
$$

where the inequality is due to (14). $\qquad \square$

*Proof of Lemma 3.5.* If $f$ is $\ell$-smooth, by Proposition 3.2, $f$ is also $(r, m)$-smooth where $m(u) = \ell(2u)$ and $r(u) = u/\ell(2u)$. Then by Lemma 3.3 where we choose $G = \|\nabla f(x)\|$, we have that $\mathcal{B}\left(x, \frac{\|\nabla f(x)\|}{\ell(2\|\nabla f(x)\|)}\right) \subseteq \mathcal{X}$, and that for any $x_1, x_2 \in \mathcal{B}\left(x, \frac{\|\nabla f(x)\|}{\ell(2\|\nabla f(x)\|)}\right)$, we have

$$
f(x_1) \leq f(x_2) + \langle \nabla f(x_2), x_1 - x_2 \rangle + \frac{\ell(2\|\nabla f(x)\|)}{2} \|x_1 - x_2\|.
$$

Choosing $x_2 = x$ and $x_1 = x - \frac{\nabla f(x)}{\ell(2\|\nabla f(x)\|)}$, it is easy to verify that $x_1, x_2 \in \mathcal{B}\left(x, \frac{\|\nabla f(x)\|}{\ell(2\|\nabla f(x)\|)}\right)$. Therefore, we have

$$
f^* \leq f\left(x - \frac{\nabla f(x)}{\ell(2\|\nabla f(x)\|)}\right) \leq f(x) - \frac{\|\nabla f(x)\|^2}{2\ell(2\|\nabla f(x)\|)},
$$

which completes the proof. $\qquad \square$

*Proof of Corollary 3.6.* We first show $G < \infty$. Note that since $\ell$ is sub-quadratic, we know $\lim_{u\to\infty} 2\ell(2u)/u^2 = 0$. Therefore, for any $F > 0$, there exists some $M > 0$ such that $2\ell(2u)/u^2 < 1/F$ for every $u > M$. In other words, for any $u$ satisfying $u^2 \leq 2\ell(2u) \cdot F$, we must have $u \leq M$. Therefore, by definition of $G$, we have $G \leq M < \infty$ if $F > 0$. If $F = 0$, we trivially get $G = 0 < \infty$. Also, since the set $\{u \geq 0 \mid u^2 \leq 2\ell(2u) \cdot F\}$ is closed and bounded, we know its supremum $G$ is in this set and it is also straightforward to show $G^2 = 2\ell(2G) \cdot F$.

Next, by Lemma 3.5, we know

$$\|\nabla f(x)\|^2 \leq 2\ell(2\|\nabla f(x)\|) \cdot (f(x) - f^*) \leq 2\ell(2\|\nabla f(x)\|) \cdot F.$$

Then based on the definition of $G$, we have $\|\nabla f(x)\| \leq G$. $\qquad\square$

# B Analysis of GD for convex functions

In this section, we provide the detailed convergence analysis of gradient descent in the convex setting, including the proofs of Lemma 4.1 and Theorem 4.2, for which the following lemma will be helpful.

**Lemma B.1** (Co-coercivity)**.** *If $f$ is convex and $(r, \ell)$-smooth, for any $x \in \mathcal{X}$ and $y \in \mathcal{B}(x, r(\|\nabla f(x)\|)/2)$, we have $y \in \mathcal{X}$ and*

$$\langle \nabla f(x) - \nabla f(y), x - y \rangle \geq \frac{1}{L} \|\nabla f(x) - \nabla f(y)\|^2,$$

*where $L = \ell(\|\nabla f(x)\|)$.*

*Proof of Lemma B.1.* Define the Bregman divergences $\phi_x(w) := f(w) - \langle \nabla f(x), w \rangle$ and $\phi_y(w) := f(w) - \langle \nabla f(y), w \rangle$, which are both convex functions. Since $\nabla \phi_x(w) = \nabla f(w) - \nabla f(x)$, we have $\nabla \phi_x(x) = 0$ which implies $\min_w \phi_x(w) = \phi_x(x)$ as $\phi_x$ is convex. Similarly we have $\min_w \phi_y(w) = \phi_y(y)$.

Denote $r_x := r(\|\nabla f(x)\|)$. Since $f$ is $(r, \ell)$-smooth, we know its gradient $\nabla f$ is $L$-Lipschitz locally in $\mathcal{B}(x, r_x)$. Since $\nabla \phi_x(w) - \nabla f(w) = \nabla f(x)$ is a constant, we know $\nabla \phi_x$ is also $L$-Lipschitz locally in $\mathcal{B}(x, r_x)$. Then similar to the proof of Lemma 3.3, one can easily show that for any $x_1, x_2 \in \mathcal{B}(x, r_x)$, we have

$$\phi_x(x_1) \leq \phi_x(x_2) + \langle \nabla \phi_x(x_2), x_1 - x_2 \rangle + \frac{L}{2} \|x_1 - x_2\|^2. \tag{15}$$

Note that for any $y \in \mathcal{B}(x, r(\|\nabla f(x)\|)/2)$ as in the lemma statement,

$$\left\| y - \frac{1}{L}\nabla \phi_x(y) - x \right\| \leq \|y - x\| + \frac{1}{L}\|\nabla f(y) - \nabla f(x)\| \leq 2\|y - x\| \leq r_x,$$

where the first inequality uses triangle inequality and $\nabla \phi_x(y) = \nabla f(y) - \nabla f(x)$; and the second inequality uses Definition 2. It implies that $y - \frac{1}{L}\nabla \phi_x(y) \in \mathcal{B}(x, r_x)$. Then we can obtain

$$\phi_x(x) = \min_w \phi_x(w) \leq \phi_x\left(y - \frac{1}{L}\nabla \phi_x(y)\right) \leq \phi_x(y) - \frac{1}{2L}\|\nabla \phi_x(y)\|^2,$$

where the last inequality uses (15) where we choose $x_1 = y - \frac{1}{L}\nabla \phi_x(y)$ and $x_2 = y$. By the definition of $\phi_x$, the above inequality is equivalent to

$$\frac{1}{2L} \|\nabla f(y) - \nabla f(x)\|^2 \leq f(y) - f(x) - \langle \nabla f(x), x - y \rangle.$$

Similar argument can be made for $\phi_y(\cdot)$ to obtain

$$\frac{1}{2L} \|\nabla f(y) - \nabla f(x)\|^2 \leq f(x) - f(y) - \langle \nabla f(y), y - x \rangle.$$

Summing up the two inequalities, we can obtain the desired result. $\qquad\square$

With Lemma B.1, we prove Lemma 4.1 as follows.

*Proof of Lemma 4.1.* Let $L = \ell(G)$. We first verify that $x^+ \in \mathcal{B}(x, r(G)/2)$. Note that

$$\left\|x^+ - x\right\| = \left\|\eta \nabla f(x)\right\| \leq \eta G \leq r(G)/2,$$

where we choose $\eta \leq r(G)/(2G)$. Thus by Lemma B.1, we have

$$
\begin{aligned}
\left\|\nabla f(x^+)\right\|^2 &= \left\|\nabla f(x)\right\|^2 + 2\langle \nabla f(x^+) - \nabla f(x), \nabla f(x)\rangle + \left\|\nabla f(x^+) - \nabla f(x)\right\|^2 \\
&= \left\|\nabla f(x)\right\|^2 - \frac{2}{\eta}\langle \nabla f(x^+) - \nabla f(x), x^+ - x\rangle + \left\|\nabla f(x^+) - \nabla f(x)\right\|^2 \\
&\leq \left\|\nabla f(x)\right\|^2 + \left(1 - \frac{2}{\eta L}\right)\left\|\nabla f(x^+) - \nabla f(x)\right\|^2 \\
&\leq \left\|\nabla f(x)\right\|^2,
\end{aligned}
$$

where the first inequality uses Lemma B.1 and the last inequality chooses $\eta \leq 2/L$. □

With Lemma 4.1, we are ready to prove both Theorem 4.2 and Theorem 4.3.

*Proof of Theorem 4.2.* Denote $G := \|\nabla f(x_0)\|$. Then we trivially have $\|\nabla f(x_0)\| \leq G$. Lemma 4.1 states that if $\|\nabla f(x_t)\| \leq G$ for any $t \geq 0$, then we also have $\|\nabla f(x_{t+1})\| \leq \|\nabla f(x_t)\| \leq G$. By induction, we can show that $\|\nabla f(x_t)\| \leq G$ for all $t \geq 0$. Then the rest of the proof basically follows the standard textbook analysis. We still provide the detailed proof below for completeness.

Note that $\|x_{t+1} - x_t\| = \eta \|\nabla f(x_t)\| \leq \eta G \leq r(G)$, where we choose $\eta \leq r(G)/(2G)$. Thus we can apply Lemma 3.3 to obtain

$$
\begin{aligned}
0 &\geq f(x_{t+1}) - f(x_t) - \langle \nabla f(x_t), x_{t+1} - x_t\rangle - \frac{L}{2}\left\|x_{t+1} - x_t\right\|^2 \\
&\geq f(x_{t+1}) - f(x_t) - \langle \nabla f(x_t), x_{t+1} - x_t\rangle - \frac{1}{2\eta}\left\|x_{t+1} - x_t\right\|^2,
\end{aligned}
\tag{16}
$$

where the last inequality chooses $\eta \leq 1/L$. Meanwhile, by convexity between $x_t$ and $x^*$, we have

$$0 \geq f(x_t) - f^* + \langle \nabla f(x_t), x^* - x_t\rangle.\tag{17}$$

Note that $(t+1)\times(16)+(17)$ gives

$$
\begin{aligned}
0 \geq{}& f(x_t) - f^* + \langle \nabla f(x_t), x^* - x_t\rangle \\
&+ (1+t)\left(f(x_{t+1}) - f(x_t) - \langle \nabla f(x_t), x_{t+1} - x_t\rangle - \frac{1}{2\eta}\left\|x_{t+1} - x_t\right\|^2\right).
\end{aligned}
$$

Then reorganizing the terms of the above inequality, noting that

$$
\begin{aligned}
\left\|x_{t+1} - x^*\right\|^2 - \left\|x_t - x^*\right\|^2 &= \left\|x_{t+1} - x_t\right\|^2 + 2\langle x_{t+1} - x_t, x_t - x^*\rangle \\
&= \left\|x_{t+1} - x_t\right\|^2 + 2\eta\langle \nabla f(x_t), x^* - x_t\rangle,
\end{aligned}
$$

we can obtain

$$(t+1)(f(x_{t+1}) - f^*) + \frac{1}{2\eta}\left\|x_{t+1} - x^*\right\|^2 \leq t(f(x_t) - f^*) + \frac{1}{2\eta}\left\|x_t - x^*\right\|^2.$$

The above inequality implies $t(f(x_t) - f^*) + \frac{1}{2\eta}\|x_t - x^*\|^2$ is a non-increasing potential function, which directly implies the desired result. □

*Proof of Theorem 4.3.* Since strongly convex functions are also convex, by the same argument as in the proof of Theorem 4.2, we have $\|\nabla f(x_t)\| \leq G := \|\nabla f(x_0)\|$ for all $t \geq 0$. Moreover, (16) still holds. For $\mu$-strongly-convex function, we can obtain a tighter version of (17) as follows.

$$0 \geq f(x_t) - f^* + \langle \nabla f(x_t), x^* - x_t\rangle + \frac{\mu}{2}\left\|x^* - x_t\right\|^2.\tag{18}$$

Let $A_0 = 0$ and $A_{t+1} = (1 + A_t)/(1 - \eta\mu)$ for all $t \geq 0$. Combining (16) and (18), we have

$$0 \geq (A_{t+1} - A_t)(f(x_t) - f^* + \langle \nabla f(x_t), x^* - x_t \rangle)$$
$$+ A_{t+1}\left( f(x_{t+1}) - f(x_t) - \langle \nabla f(x_t), x_{t+1} - x_t \rangle - \frac{1}{2\eta} \|x_{t+1} - x_t\|^2 \right).$$

Then reorganizing the terms of the above inequality, noting that

$$\|x_{t+1} - x^*\|^2 - \|x_t - x^*\|^2 = \|x_{t+1} - x_t\|^2 + 2\langle x_{t+1} - x_t, x_t - x^* \rangle$$
$$= \|x_{t+1} - x_t\|^2 + 2\eta\langle \nabla f(x_t), x^* - x_t \rangle,$$

we can obtain

$$A_{t+1}(f(x_{t+1}) - f^*) + \frac{1 + \eta\mu A_{t+1}}{2\eta} \|x_{t+1} - x^*\|^2 \leq A_t(f(x_t) - f^*) + \frac{1 + \eta\mu A_t}{2\eta} \|x_t - x^*\|^2.$$

The above inequality means $A_t(f(x_t) - f^*) + \frac{1+\eta\mu A_t}{2\eta} \|x_t - x^*\|^2$ is a non-increasing potential function. Thus by telescoping we have

$$f(x_T) - f^* \leq \frac{\mu(1 - \eta\mu)^T}{2(1 - (1 - \eta\mu)^T)} \|x_0 - x^*\|^2.$$

$\square$

## C  Analysis of NAG for convex functions

In this section, we provide the detailed analysis of Nesterov's accelerated gradient method in the convex setting. As we discussed in Section 4.2, the stepsize size choice in Theorem 4.4 is smaller than the classical one. Therefore, we provide a more fine-grained version of the theorem, which allows the stepsize to depend on the degree of $\ell$.

**Theorem C.1.** *Suppose $f$ is convex and $\ell$-smooth. For $\alpha \in (0, 2]$, if $\ell(u) = o(u^\alpha)$, i.e., $\lim_{u \to \infty} \ell(u)/u^\alpha = 0$, then there must exist a constant $G$ such that for $L := \ell(2G)$, we have*

$$G \geq \max\left\{ 8\max\{L^{1/\alpha - 1/2}, 1\}\sqrt{L((f(x_0) - f^*) + \|x_0 - x^*\|^2)}, \|\nabla f(x_0)\| \right\}. \quad (19)$$

*Choose $\eta \leq \min\left\{ \frac{1}{16L^{3-2/\alpha}}, \frac{1}{2L} \right\}$. Then the iterates of Algorithm 1 satisfy*

$$f(x_T) - f^* \leq \frac{4(f(x_0 - f^*) + 4\|x_0 - x^*\|^2}{\eta T^2 + 4}.$$

Note that when $\alpha = 2$, i.e., $\ell$ is sub-quadratic, Theorem C.1 reduces to Theorem 4.4 which chooses $\eta \leq \min\{\frac{1}{16L^2}, \frac{1}{2L}\}$. When $\alpha = 1$, i.e., $\ell$ is sub-linear, the above theorem chooses $\eta \leq \frac{1}{16L}$ as in the classical textbook analysis up to a numerical constant factor.

Throughout this section, we will assume $f$ is convex and $\ell$-smooth, and consider the parameter choices in Theorem C.1, unless explicitly stated. Note that since $f$ is $\ell$-smooth, it is also $(r, m)$-smooth with $m(u) = \ell(u + G)$ and $r(u) = \frac{G}{\ell(u+G)}$ by Proposition 3.2. Note that $m(G) = \ell(2G) = L$ and $r(G) = G/L$. Then the stepsize satisfies $\eta \leq 1/(2L) \leq \min\{\frac{2}{m(G)}, \frac{r(G)}{2G}\}$.

Before proving Theorem C.1, we first present several additional useful lemmas. To start with, we provide two lemmas regarding the weights $\{A_t\}_{t \geq 0}$ and $\{B_t\}_{t \geq 0}$ used in Algorithm 1. The lemma below states that $B_t = \Theta(t^2)$.

**Lemma C.2.** *The weights $\{B_t\}_{t \geq 0}$ in Algorithm 1 satisfy $\frac{1}{4}t^2 \leq B_t \leq t^2$ for all $t \geq 0$.*

*Proof of Lemma C.2.* We prove this lemma by induction. First note that the inequality obviously holds for $B_0 = 0$. Suppose its holds up to $t$. Then we have

$$B_{t+1} = B_t + \frac{1}{2}(1 + \sqrt{4B_t + 1}) \geq \frac{1}{4}t^2 + \frac{1}{2}(1 + \sqrt{t^2 + 1}) \geq \frac{1}{4}(t + 1)^2.$$

Similarly, we have

$$B_{t+1} = B_t + \frac{1}{2}(1 + \sqrt{4B_t + 1}) \le t^2 + \frac{1}{2}(1 + \sqrt{4t^2 + 1}) \le (t+1)^2.$$

$\square$

Lemma C.2 implies the following useful lemma.

**Lemma C.3.** *The weights $\{A_t\}_{t \ge 0}$ in Algorithm 1 satisfy that*

$$(1 - \frac{A_t}{A_{t+1}}) \frac{1}{A_t} \sum_{s=0}^{t-1} \sqrt{A_{s+1}} (A_{s+1} - A_s - 1) \le 4.$$

*Proof of Lemma C.3.* First, note that it is easy to verify that $A_{s+1} - A_s - 1 = B_{s+1} - B_s - 1 \ge 0$, which implies each term in the LHS of the above inequality is non-negative. Then we have

$$(1 - \frac{A_t}{A_{t+1}}) \frac{1}{A_t} \sum_{s=0}^{t-1} \sqrt{A_{s+1}} (A_{s+1} - A_s - 1)$$

$$\le \frac{1}{A_{t+1}\sqrt{A_t}} (A_{t+1} - A_t) \sum_{s=0}^{t-1} (A_{s+1} - A_s - 1) \qquad (A_t \ge A_{s+1})$$

$$= \frac{1}{A_{t+1}\sqrt{A_t}} (B_{t+1} - B_t) \sum_{s=0}^{t-1} (B_{s+1} - B_s - 1) \qquad (A_s = B_s + 1/\eta)$$

$$= \frac{1}{A_{t+1}\sqrt{A_t}} \cdot \frac{1}{2}(1 + \sqrt{4B_t + 1}) \sum_{s=0}^{t-1} \left( -1 + \frac{1}{2}(1 + \sqrt{4B_s + 1}) \right) \qquad (\text{by definition of } B_s)$$

$$\le 8 \frac{1}{(t+1)^2 t} \cdot (t+1) \frac{t^2}{2} \qquad (\text{by } A_t \ge B_t \text{ and Lemma C.2})$$

$$\le 4.$$

$\square$

The following lemma summarizes the results in the classical potential function analysis of NAG in [d'Aspremont et al., 2021]. In order to not deal with the generalized smoothness condition for now, we directly assume the inequality (20) holds in the lemma, which will be proved later under the generalized smoothness condition.

**Lemma C.4.** *For any $t \ge 0$, if the following inequality holds,*

$$f(y_t) + \langle \nabla f(y_t), x_{t+1} - y_t \rangle + \frac{1}{2\eta} \|x_{t+1} - y_t\|^2 \ge f(x_{t+1}), \tag{20}$$

*then we can obtain*

$$A_{t+1}(f(x_{t+1}) - f^*) + \frac{1}{2\eta} \|z_{t+1} - x^*\|^2 \le A_t(f(x_t) - f^*) + \frac{1}{2\eta} \|z_t - x^*\|^2. \tag{21}$$

*Proof of Lemma C.4.* These derivations below can be found in [d'Aspremont et al., 2021]. We present them here for completeness.

First, since $f$ is convex, the convexity between $x^*$ and $y_t$ gives

$$f^* \ge f(y_t) + \langle \nabla f(y_t), x^* - y_t \rangle.$$

Similarly the convexity between $x_t$ and $y_t$ gives

$$f(x_t) \ge f(y_t) + \langle \nabla f(y_t), x_t - y_t \rangle.$$

Combining the above two inequalities as well as (20) assumed in this lemma, we have

$$
\begin{aligned}
0 \geq\ & (A_{t+1} - A_t)(f(y_t) - f^* + \langle \nabla f(y_t), x^* - y_t \rangle) \\
& + A_t(f(y_t) - f(x_t) + \langle \nabla f(y_t), x_t - y_t \rangle) \\
& + A_{t+1}\left( f(x_{t+1}) - f(y_t) - \langle \nabla f(y_t), x_{t+1} - y_t \rangle - \frac{1}{2\eta} \|x_{t+1} - y_t\|^2 \right).
\end{aligned}
\tag{22}
$$

Furthermore, note that

$$
\begin{aligned}
& \frac{1}{2\eta}\left( \|z_{t+1} - x^*\|^2 - \|z_t - x^*\|^2 \right) \\
&= \frac{1}{2\eta}\left( \|z_{t+1} - z_t\|^2 + 2\langle z_{t+1} - z_t, z_t - x^* \rangle \right) \\
&= \frac{1}{2\eta}\left( \eta^2 (A_{t+1} - A_t)^2 \|\nabla f(y_t)\|^2 - 2\eta(A_{t+1} - A_t)\langle \nabla f(y_t), z_t - x^* \rangle \right) \\
&= \frac{\eta}{2}(A_{t+1} - A_t)^2 \|\nabla f(y_t)\|^2 - (A_{t+1} - A_t)\langle \nabla f(y_t), z_t - x^* \rangle.
\end{aligned}
\tag{23}
$$

Meanwhile, we have

$$
A_{t+1}x_{t+1} = A_{t+1}y_t - \eta A_{t+1}\nabla f(y_t) = A_{t+1}x_t + (A_{t+1} - A_t)(z_t - x_t) - \eta A_{t+1}\nabla f(y_t).
$$

Thus we have

$$
(A_{t+1} - A_t)z_t = A_{t+1}x_{t+1} - A_t x_t + \eta A_{t+1}\nabla f(y_t).
$$

Plugging back in (23), we obtain

$$
\begin{aligned}
& \frac{1}{2\eta}\left( \|z_{t+1} - x^*\|^2 - \|z_t - x^*\|^2 \right) \\
&= \frac{\eta}{2}(A_{t+1} - A_t)^2 \|\nabla f(y_t)\|^2 + (A_{t+1} - A_t)\langle \nabla f(y_t), x^* \rangle \\
&\quad + \langle -A_{t+1}x_{t+1} + A_t x_t - \eta A_{t+1}\nabla f(y_t), \nabla f(y_t) \rangle.
\end{aligned}
$$

Thus

$$
\begin{aligned}
& (A_{t+1} - A_t)\langle \nabla f(y_t), x^* \rangle + \langle A_t x_t - A_{t+1}x_{t+1}, \nabla f(y_t) \rangle \\
&= \frac{1}{2\eta}\left( \|z_{t+1} - x^*\|^2 - \|z_t - x^*\|^2 \right) + \eta(A_{t+1} - \tfrac{1}{2}(A_{t+1} - A_t)^2)\|\nabla f(y_t)\|^2.
\end{aligned}
$$

So we can reorganize (22) to obtain

$$
\begin{aligned}
0 \geq\ & A_{t+1}(f(x_{t+1}) - f^*) - A_t(f(x_t) - f^*) \\
& + (A_{t+1} - A_t)\langle \nabla f(y_t), x^* \rangle + \langle A_t x_t - A_{t+1}x_{t+1}, \nabla f(y_t) \rangle \\
& - \frac{1}{2\eta}A_{t+1}\|x_{t+1} - y_t\|^2 \\
=\ & A_{t+1}(f(x_{t+1}) - f^*) - A_t(f(x_t) - f^*) \\
& + \frac{1}{2\eta}\left( \|z_{t+1} - x^*\|^2 - \|z_t - x^*\|^2 \right) + \frac{\eta}{2}(A_{t+1} - (A_{t+1} - A_t)^2)\|\nabla f(y_t)\|^2.
\end{aligned}
$$

Then we complete the proof noting that it is easy to verify

$$
A_{t+1} - (A_{t+1} - A_t)^2 = B_{t+1} + \frac{1}{\eta} - (B_{t+1} - B_t)^2 = \frac{1}{\eta} \geq 0.
$$

$\square$

In the next lemma, we show that if $\|\nabla f(y_t)\| \leq G$, then the condition (20) assumed in Lemma C.4 is satisfied at time $t$.

**Lemma C.5.** *For any $t \geq 0$, if $\|\nabla f(y_t)\| \leq G$, then we have $\|\nabla f(x_{t+1})\| \leq G$, and furthermore,*

$$
f(y_t) + \langle \nabla f(y_t), x_{t+1} - y_t \rangle + \frac{1}{2\eta} \|x_{t+1} - y_t\|^2 \geq f(x_{t+1}).
$$

*Proof of Lemma C.5.* As disccued below Theorem C.1, the stepsize satisfies $\eta \leq 1/(2L) \leq \min\{\frac{2}{m(G)}, \frac{r(G)}{2G}\}$. Therefore we can apply Lemma 4.1 to show $\|\nabla f(x_{t+1})\| \leq \|\nabla f(y_t)\| \leq G$. For the second part, note that $\|x_{t+1} - y_t\| = \eta \|\nabla f(y_t)\| \leq \frac{G}{2L} \leq r(G)$, we can apply Lemma 3.3 to show

$$f(x_{t+1}) \leq f(y_t) + \langle \nabla f(y_t), x_{t+1} - y_t \rangle + \frac{L}{2} \|x_{t+1} - y_t\|^2$$
$$\leq f(y_t) + \langle \nabla f(y_t), x_{t+1} - y_t \rangle + \frac{1}{2\eta} \|x_{t+1} - y_t\|^2.$$

$\square$

With Lemma C.4 and Lemma C.5, we can show that $\|\nabla f(y_t)\| \leq G$ for all $t \geq 0$, as in the lemma below.

**Lemma C.6.** *For all $t \geq 0$, $\|\nabla f(y_t)\| \leq G$.*

*Proof of Lemma C.6.* We will prove this lemma by induction. First, by Lemma 3.5 and the choice of $G$, it is easy to verify that $\|\nabla f(x_0)\| \leq G$. Then for any fixed $t \geq 0$, suppose that $\|\nabla f(x_s)\| \leq G$ for all $s < t$. Then by Lemma C.4 and Lemma C.5, we know that $\|\nabla f(x_s)\| \leq G$ for all $0 \leq s \leq t$, and that for all $s < t$,

$$A_{s+1}(f(x_{s+1}) - f^*) + \frac{1}{2\eta} \|z_{s+1} - x^*\|^2 \leq A_s(f(x_s) - f^*) + \frac{1}{2\eta} \|z_s - x^*\|^2. \tag{24}$$

By telescoping (24), we have for all $0 \leq s < t$,

$$f(x_{s+1}) - f^* \leq \frac{1}{\eta A_{s+1}}((f(x_0) - f^*) + \|z_0 - x^*\|^2). \tag{25}$$

For $0 \leq s \leq t$, since $\|\nabla f(x_s)\| \leq G$, then Lemma 3.5 implies

$$\|\nabla f(x_s)\|^2 \leq 2L(f(x_s) - f^*). \tag{26}$$

Note that by Algorithm 1, we have

$$z_t - x_t = \frac{A_{t-1}}{A_t}(z_{t-1} - x_{t-1}) - \eta(A_t - A_{t-1})\nabla f(y_{t-1}) + \eta \nabla f(y_{t-1}).$$

Thus we can obtain

$$z_t - x_t = -\frac{1}{A_t} \sum_{s=1}^{t-1} \eta A_{s+1}(A_{s+1} - A_s - 1)\nabla f(y_s).$$

Therefore

$$y_t - x_t = -(1 - \frac{A_t}{A_{t+1}})\frac{1}{A_t} \sum_{s=1}^{t-1} \eta A_{s+1}(A_{s+1} - A_s - 1)\nabla f(y_s).$$

Thus we have

$$\|y_t - x_t\| \leq (1 - \frac{A_t}{A_{t+1}})\frac{1}{A_t} \sum_{s=1}^{t-1} \eta A_{s+1}(A_{s+1} - A_s - 1) \|\nabla f(y_s)\| =: \mathcal{I}.$$

Since $\|\nabla f(y_s)\| \leq G$ and $\|x_{s+1} - y_s\| = \|\eta \nabla f(y_s)\| \leq r(G)$ for $s < t$, by Lemma 3.3, we have

$$\mathcal{I} \leq (1 - \frac{A_t}{A_{t+1}})\frac{1}{A_t} \sum_{s=1}^{t-1} \eta A_{s+1}(A_{s+1} - A_s - 1)(\|\nabla f(x_{s+1})\| + \eta L \|\nabla f(y_s)\|)$$

$$\leq \eta L \mathcal{I} + (1 - \frac{A_t}{A_{t+1}})\frac{1}{A_t} \sum_{s=1}^{t-1} \eta A_{s+1}(A_{s+1} - A_s - 1) \|\nabla f(x_{s+1})\|.$$

Thus

$$\|y_t - x_t\|$$

$$\leq \mathcal{I} \leq \frac{1}{1-\eta L}\left(1 - \frac{A_t}{A_{t+1}}\right)\frac{1}{A_t}\sum_{s=1}^{t-1}\eta A_{s+1}(A_{s+1}-A_s-1)\|\nabla f(x_{s+1})\|$$

$$\leq \frac{1}{1-\eta L}\left(1 - \frac{A_t}{A_{t+1}}\right)\frac{1}{A_t}\sum_{s=1}^{t-1}\eta A_{s+1}(A_{s+1}-A_s-1)\sqrt{2L(f(x_{s+1})-f^*)} \qquad \text{(by (26))}$$

$$\leq \frac{1}{1-\eta L}\left(1 - \frac{A_t}{A_{t+1}}\right)\frac{1}{A_t}\sum_{s=1}^{t-1}\eta A_{s+1}(A_{s+1}-A_s-1)\sqrt{\frac{2L}{A_{s+1}}\cdot\frac{1}{\eta}((f(x_0)-f^*)+\|z_0-x^*\|^2)}$$

$$\text{(by (25))}$$

$$= \frac{2\sqrt{\eta L}}{1-\eta L}\left(1 - \frac{A_t}{A_{t+1}}\right)\frac{1}{A_t}\sum_{s=1}^{t-1}\sqrt{A_{s+1}}(A_{s+1}-A_s-1)\sqrt{(f(x_0)-f^*)+\|z_0-x^*\|^2}$$

$$\leq \frac{8\sqrt{\eta}}{1-\eta L}\sqrt{L((f(x_0)-f^*)+\|z_0-x^*\|^2)} \qquad \text{(by Lemma C.3)}$$

$$\leq \frac{1}{2L^{3/2-1/\alpha}}\cdot L^{1/2-1/\alpha}G = \frac{G}{2L} \leq r(G). \qquad \text{(by the choices of $\eta$ and $G$)}$$

Since $\|\nabla f(x_t)\| \leq G$ and we just showed $\|x_t - y_t\| \leq r(G)$, by Lemma 3.3, we have

$$\|\nabla f(y_t)\| \leq \|\nabla f(x_t)\| + L\|y_t - x_t\|$$

$$\leq \sqrt{\frac{2L}{\eta A_t}((f(x_0)-f^*)+\|z_0-x^*\|^2)} + L\cdot\frac{G}{2L} \qquad \text{(by (26) and (25))}$$

$$\leq G\left(\frac{1}{4}+\frac{1}{2}\right) \leq G. \qquad \text{(by $A_t \geq 1/\eta$ and choice of $G$)}$$

Then we complete the induction as well as the proof.

$\square$

With the three lemmas above, it is straight forward to prove Theorem C.1.

*Proof of Theorem C.1.* Combining Lemmas C.4, C.5, and C.6, we know the following inequality holds for all $t \geq 0$.

$$A_{t+1}(f(x_{t+1})-f^*) + \frac{1}{2\eta}\|z_{t+1}-x^*\|^2 \leq A_t(f(x_t)-f^*) + \frac{1}{2\eta}\|z_t-x^*\|^2,$$

Then by telescoping, we directly complete the proof.

$\square$

# D   Analysis of NAG for strongly convex functions

In this section, we provide the convergence analysis of the modified version of Nesterov's accelerated gradient method for $\mu$-strongly-convex functions defined in Algorithm 2.

The convergence results is formally presented in the following theorem.

**Theorem D.1.** *Suppose $f$ is $\mu$-strongly-convex and $\ell$-smooth. For $\alpha \in (0,2]$, if $\ell(u) = o(u^\alpha)$, i.e., $\lim_{u\to\infty}\ell(u)/u^\alpha = 0$, then there must exist a constant $G$ such that for $L := \ell(2G)$, we have*

$$G \geq 8\max\{L^{1/\alpha-1/2},1\}\sqrt{L((f(x_0)-f^*)+\mu\|z_0-x^*\|^2)/\min\{\mu,1\}}. \qquad (27)$$

*If we choose*

$$\eta \leq \min\left\{\frac{1}{144L^{3-2/\alpha}\log^4\left(e+\frac{144L^{3-2/\alpha}}{\mu}\right)}, \frac{1}{2L}\right\}. \qquad (28)$$

---

Algorithm 2: NAG for $\mu$-strongly-convex functions

---

**input** A $\mu$-strongly-convex and $\ell$-smooth function $f$, stepsize $\eta$, initial point $x_0$

1: **Initialize** $z_0 = x_0$, $B_0 = 0$, and $A_0 = 1/(\eta\mu)$.
2: **for** $t = 0, \ldots$ **do**
3:     $B_{t+1} = \frac{2B_t + 1 + \sqrt{4B_t + 4\eta\mu B_t^2 + 1}}{2(1 - \eta\mu)}$
4:     $A_{t+1} = B_{t+1} + \frac{1}{\eta\mu}$
5:     $\tau_t = \frac{(A_{t+1} - A_t)(1 + \eta\mu A_t)}{A_{t+1} + 2\eta\mu A_t A_{t+1} - \eta\mu A_t^2}$ and $\delta_t = \frac{A_{t+1} - A_t}{1 + \eta\mu A_{t+1}}$
6:     $y_t = x_t + \tau_t(z_t - x_t)$
7:     $x_{t+1} = y_t - \eta\nabla f(y_t)$
8:     $z_{t+1} = (1 - \eta\mu\delta_t)z_t + \eta\mu\delta_t y_t - \eta\delta_t\nabla f(y_t)$
9: **end for**

---

*The iterates generated by Algorithm 2 satisfy*

$$f(x_T) - f^* \leq \frac{(1 - \sqrt{\eta\mu})^{T-1}(f(x_0 - f^*) + \mu\|z_0 - x^*\|^2)}{\eta\mu + (1 - \sqrt{\eta\mu})^{T-1}}.$$

The above theorem gives a gradient complexity of $\mathcal{O}\left(\frac{1}{\sqrt{\eta\mu}}\log(1/\epsilon)\right)$. Note that Theorem 4.2 shows the complexity of GD is $\mathcal{O}\left(\frac{1}{\eta\mu}\log(1/\epsilon)\right)$. It seems NAG gives a better rate at first glance. However, note that the choices of $G, L, \eta$ in these two theorems are different, it is less clear whether NAG accelerates the optimization in this setting. Below, we informally show that, if $\ell(u) = o(\sqrt{u})$, the rate we obtain for NAG is faster than that for GD.

For simplicity, we informally assume $\ell(u) \asymp x^\rho$ with $\rho \in (0, 1)$. Let $G_0 = \|\nabla f(x_0)\|$. Then for GD, by Theorem 4.2, we have $\eta_{\mathrm{gd}}\mu \asymp \mu/\ell(G_0) \asymp \mu/G_0^\rho$. For NAG, since $\ell$ is sub-linear we can choose $\alpha = 1$ in the theorem statement. Since $f$ is $\mu$-strongly-convex, by standard results, we can show that $f(x_0) - f^* \leq \frac{1}{\mu}G_0^2$ and $\|z_0 - x^*\| \leq \frac{1}{\mu}G_0$. Thus the requirement of $G$ in (27) can be simplified as $G \gtrsim \ell(G) \cdot G_0/\mu$, which is satisfied if choosing $G \asymp (G_0/\mu)^{1/(1-\rho)}$. Then we also have $\eta_{\mathrm{nag}} \asymp \frac{1}{\ell(G)} \asymp (\mu/G_0)^{\rho/(1-\rho)}$. Thus $\sqrt{\eta_{\mathrm{nag}}\mu} \asymp (\mu/G_0^\rho)^{1/(2-2\rho)}$. This means whenever $1/(2 - 2\rho) < 1$, i.e., $0 \leq \rho < 1/2$, we have $\sqrt{\eta_{\mathrm{nag}}\mu} \gtrsim \eta_{\mathrm{gd}}\mu$, which implies the rate we obtain for NAG is faster than that for GD.

In what follows, we will provide the proof of Theorem D.1. We will always use the parameter choices in the theorem throughout this section.

### D.1 Useful lemmas

In this part, we provide several useful lemmas for proving Theorem D.1. To start with, the following two lemmas provide two useful inequalities.

**Lemma D.2.** *For any $0 \leq u \leq 1$, we have $\log(1 + u) \geq \frac{1}{2}u$.*

**Lemma D.3.** *For all $0 < p \leq 1$ and $t \geq 0$, we have*

$$t \leq \frac{2}{\sqrt{p}}\log(e + \frac{1}{p})(p(1 + \sqrt{p})^t + 1).$$

*Proof of Lemma D.3.* Let

$$f(t) = \frac{2}{\sqrt{p}}\log(e + \frac{1}{p})(p(1 + \sqrt{p})^t + 1) - t.$$

It is obvious that $f(t) \geq 0$ for $t \leq \frac{2}{\sqrt{p}} \log(e + \frac{1}{p})$. For $t > \frac{2}{\sqrt{p}} \log(e + \frac{1}{p})$, we have

$$
\begin{aligned}
f'(t) &= 2\sqrt{p} \log(e + \frac{1}{p}) \log(1 + \sqrt{p})(1 + \sqrt{p})^t - 1 \\
&\geq p(1 + \sqrt{p})^t - 1 && \text{(by Lemma D.2)} \\
&= p \exp(t \log(1 + \sqrt{p})) - 1 \\
&\geq p \exp(t\sqrt{p}/2) - 1 && \text{(by Lemma D.2)} \\
&\geq p(e + 1/p) - 1 \geq 0. && \text{(since } t > \frac{2}{\sqrt{p}} \log(e + \frac{1}{p}))
\end{aligned}
$$

Thus $f$ is non-decreasing and

$$
f(t) \geq f\left(\frac{2}{\sqrt{p}} \log(e + \frac{1}{p})\right) \geq 0.
$$

$\square$

In the next four lemmas, we provide several useful inequalities regarding the weights $\{A_t\}_{t \geq 0}$ and $\{B_t\}_{t \geq 0}$ used in Algorithm 2.

**Lemma D.4.** *For all $s \leq t$, we have*

$$
\frac{B_{t+1} - B_t}{B_{t+1}} \cdot \frac{B_{s+1} - B_s}{1 + \eta\mu B_{s+1}} \leq 1,
$$

*which implies $\tau_t \cdot \delta_s \leq 1$.*

*Proof of Lemma D.4.* By Algorithm 2, it is easy to verify

$$
(B_{s+1} - B_s)^2 = B_{s+1}(1 + \eta\mu B_{s+1}).
$$

This implies

$$
B_s = B_{s+1} - \sqrt{B_{s+1}(1 + \eta\mu B_{s+1})}.
$$

Thus

$$
\frac{B_t}{B_{t+1}} = 1 - \sqrt{\eta\mu + \frac{1}{B_{t+1}}} \geq 1 - \sqrt{\eta\mu + \frac{1}{B_{s+1}}} = \frac{B_s}{B_{s+1}},
$$

where in the inequality, we use the fact that $B_s$ is non-decreasing with $s$. Therefore

$$
\frac{B_{t+1} - B_t}{B_{t+1}} \cdot \frac{B_{s+1} - B_s}{1 + \eta\mu B_{s+1}} \leq \frac{B_{s+1} - B_s}{B_{s+1}} \cdot \frac{B_{s+1} - B_s}{1 + \eta\mu B_{s+1}} = 1.
$$

Thus we have

$$
\begin{aligned}
\tau_t \cdot \delta_s &= \frac{(A_{t+1} - A_t)(1 + \eta\mu A_t)}{A_{t+1} + 2\eta\mu A_t A_{t+1} - \eta\mu A_t^2} \cdot \frac{A_{s+1} - A_s}{1 + \eta\mu A_{s+1}} \\
&\leq \frac{A_{t+1} - A_t}{A_{t+1}} \cdot \frac{A_{s+1} - A_s}{1 + \eta\mu A_{s+1}} && \text{(by } A_{t+1} \geq A_t) \\
&= \frac{B_{t+1} - B_t}{A_{t+1}} \cdot \frac{B_{s+1} - B_s}{1 + \eta\mu A_{s+1}} && \text{(by } A_{s+1} - A_s = B_{s+1} - B_s) \\
&\leq \frac{B_{t+1} - B_t}{B_{t+1}} \cdot \frac{B_{s+1} - B_s}{1 + \eta\mu B_{s+1}} \leq 1. && \text{(by } A_{s+1} \geq B_{s+1})
\end{aligned}
$$

$\square$

**Lemma D.5.** *If $0 < \eta\mu < 1$, then for any $t \geq 1$, we have*

$$
\frac{B_t}{1 - \sqrt{\eta\mu}} \leq B_{t+1} \leq \frac{3B_t}{1 - \eta\mu}.
$$

*Thus*

$$
B_t \geq \frac{1}{(1 - \sqrt{\eta\mu})^{t-1}} \geq (1 + \sqrt{\eta\mu})^{t-1}.
$$

*Proof of Lemma D.5.* For $t \geq 1$, we have $B_t \geq 1$ thus

$$B_{t+1} = \frac{2B_t + 1 + \sqrt{4B_t + 4\eta\mu B_t^2 + 1}}{2(1 - \eta\mu)} \leq \frac{2B_t + 1}{1 - \eta\mu} \leq \frac{3B_t}{1 - \mu\eta}.$$

On the other hand, we have

$$\begin{aligned}
B_{t+1} &= \frac{2B_t + 1 + \sqrt{4B_t + 4\eta\mu B_t^2 + 1}}{2(1 - \eta\mu)} \\
&\geq \frac{2B_t + \sqrt{(2B_t\sqrt{\eta\mu})^2}}{2(1 - \eta\mu)} \\
&= \frac{B_t}{1 - \sqrt{\eta\mu}}.
\end{aligned}$$

Thus

$$B_t \geq \left(\frac{1}{1 - \sqrt{\eta\mu}}\right)^{t-1} B_1 \geq \left(\frac{1}{1 - \sqrt{\eta\mu}}\right)^{t-1} \geq (1 + \sqrt{\eta\mu})^{t-1}.$$

$\square$

**Lemma D.6.** *For $0 < \eta\mu < 1$ and $t \geq 1$, we have*

$$\sum_{s=0}^{t} \sqrt{B_s} \leq (1 - \eta\mu)B_{t+1} \leq 3B_t.$$

*Proof of Lemma D.6.*

$$\begin{aligned}
B_{t+1} &= \frac{2B_t + 1 + \sqrt{4B_t + 4\eta\mu B_t^2 + 1}}{2(1 - \eta\mu)} \\
&\geq B_t + \frac{\sqrt{B_t}}{1 - \eta\mu} \\
&\geq \cdots \\
&\geq \sum_{s=0}^{t} \frac{\sqrt{B_s}}{1 - \eta\mu}.
\end{aligned}$$

Combined with Lemma D.5, we have the desired result. $\square$

**Lemma D.7.** *For $t \geq 1$, we have*

$$\sum_{s=0}^{t-1} \frac{\sqrt{A_{s+1}}}{A_t} \leq 3 + 4\log(e + \frac{1}{\eta\mu}).$$

*Proof of Lemma D.7.* By Lemma D.5, we have

$$A_t = B_t + \frac{1}{\eta\mu} \geq (1 + \sqrt{\eta\mu})^{t-1} + \frac{1}{\eta\mu}. \tag{29}$$

Thus, we have

$$\begin{aligned}
\sum_{s=0}^{t-1} \frac{\sqrt{A_{s+1}}}{A_t} &= \sum_{s=0}^{t-1} \frac{\sqrt{B_{s+1} + 1/(\eta\mu)}}{A_t} \\
&\leq \sum_{s=0}^{t-1} \frac{\sqrt{B_{s+1}}}{A_t} + \frac{t}{\sqrt{\eta\mu}A_t} \\
&\leq 3 + \frac{1}{\sqrt{\eta\mu}A_t} \cdot \frac{2}{\sqrt{\eta\mu}} \log(e + \frac{1}{\eta\mu})(\eta\mu(1 + \sqrt{\eta\mu})^t + 1) \\
&\hspace{4cm} \text{(by Lemma D.6 and Lemma D.3)} \\
&\leq 3 + 4\log(e + \frac{1}{\eta\mu}). \hspace{2cm} \text{(by Inequality (29))}
\end{aligned}$$

$\square$

## D.2 Proof of Theorem D.1

With all the useful lemmas in the previous section, we proceed to prove Theorem D.1, for which we need several additional lemmas. First, similar to Lemma C.4, the following lemma summarizes the results in the classical potential function analysis of NAG for strongly convex functions in [d'Aspremont et al., 2021].

**Lemma D.8.** *For any $t \geq 0$, if the following inequality holds*

$$f(y_t) + \langle \nabla f(y_t), x_{t+1} - y_t \rangle + \frac{1}{2\eta} \|x_{t+1} - y_t\|^2 \geq f(x_{t+1}),$$

*then we can obtain*

$$A_{t+1}(f(x_{t+1}) - f^*) + \frac{1 + \eta\mu A_{t+1}}{2\eta} \|z_{t+1} - x^*\|^2 \leq A_t(f(x_t) - f^*) + \frac{1 + \eta\mu A_t}{2\eta} \|z_t - x^*\|^2.$$

*Proof of Lemma D.8.* These derivations can be found in d'Aspremont et al. [2021]. We present it here for completeness.

The strong convexity between $x^*$ and $y_t$ gives

$$f^* \geq f(y_t) + \langle \nabla f(y_t), x^* - y_t \rangle + \frac{\mu}{2} \|x^* - y_t\|^2.$$

The convexity between $x_t$ and $y_t$ gives

$$f(x_t) \geq f(y_t) + \langle \nabla f(y_t), x_t - y_t \rangle.$$

Combining the above two inequalities and the one assumed in this lemma, we have

$$\begin{aligned}
0 \geq\ & (A_{t+1} - A_t)(f^* - f(y_t) - \langle \nabla f(y_t), x^* - y_t \rangle - \frac{\mu}{2} \|x^* - y_t\|^2) \\
& + A_t(f(y_t) - f(x_t) - \langle \nabla f(y_t), x_t - y_t \rangle) \\
& + A_{t+1}(f(x_{t+1}) - f(y_t) - \langle \nabla f(y_t), x_{t+1} - y_t \rangle - \frac{1}{2\eta} \|x_{t+1} - y_t\|^2).
\end{aligned}$$

Reorganizing we can obtain

$$\begin{aligned}
& A_{t+1}(f(x_{t+1}) - f^*) + \frac{1 + \eta\mu A_{t+1}}{2\eta} \|z_{t+1} - x^*\|^2 \\
& \leq A_t(f(x_t) - f^*) + \frac{1 + \eta\mu A_t}{2\eta} \|z_t - x^*\|^2 \\
& \quad + \frac{(A_t - A_{t+1})^2 - A_{t+1} - \eta\mu A_{t+1}^2}{1 + \eta\mu A_{t+1}} \frac{\eta}{2} \|\nabla f(y_t)\|^2 \\
& \quad - A_t^2 \frac{(A_{t+1} - A_t)(1 + \eta\mu A_t)(1 + \eta\mu A_{t+1})}{(A_{t+1} + 2\eta\mu A_t A_{t+1} - \eta\mu A_t^2)^2} \frac{\mu}{2} \|x_t - z_t\|^2.
\end{aligned}$$

Then we complete the proof noting that

$$\begin{aligned}
& (A_t - A_{t+1})^2 - A_{t+1} - \eta\mu A_{t+1}^2 \\
& = (B_t - B_{t+1})^2 - B_{t+1} + \frac{1}{\eta\mu} - \eta\mu(B_{t+1} + 1/(\eta\mu))^2 \\
& = \eta\mu B_{t+1}^2 + \frac{1}{\eta\mu} - \eta\mu B_{t+1}^2 - 2B_{t+1} - \frac{1}{\eta\mu} \\
& = -2B_{t+1} \leq 0.
\end{aligned}$$

$\square$

Next, note that Lemma C.5 still holds in the strongly convex setting. We repeat it below for completeness.

**Lemma D.9.** *For any $t \geq 0$, if $\|\nabla f(y_t)\| \leq G$, then we have $\|\nabla f(x_{t+1})\| \leq G$, and furthermore,*

$$f(y_t) + \langle \nabla f(y_t), x_{t+1} - y_t \rangle + \frac{1}{2\eta} \|x_{t+1} - y_t\|^2 \geq f(x_{t+1}).$$

With Lemma D.8 and Lemma D.9, we will show that $\|\nabla f(y_t)\| \leq G$ for all $t \geq 0$ by induction in the following lemma.

**Lemma D.10.** *For all $t \geq 0$, we have $\|\nabla f(y_t)\| \leq G$.*

*Proof of Lemma D.10.* We will prove this lemma by induction. First, by Lemma 3.5 and the choice of $G$, it is easy to verify that $\|\nabla f(x_0)\| \leq G$. Then for any fixed $t \geq 0$, suppose that $\|\nabla f(x_s)\| \leq G$ for all $s < t$. Then by Lemma D.8 and Lemma D.9, we know that $\|\nabla f(x_s)\| \leq G$ for all $0 \leq s \leq t$, and that for all $s < t$,

$$A_{s+1}(f(x_{s+1}) - f^*) + \frac{1 + \eta\mu A_{s+1}}{2\eta} \|z_{s+1} - x^*\|^2 \leq A_s(f(x_s) - f^*) + \frac{1 + \eta\mu A_s}{2\eta} \|z_s - x^*\|^2. \tag{30}$$

By telescoping (30), we have for all $0 \leq s < t$,

$$f(x_{s+1}) - f^* \leq \frac{1}{A_{s+1}\eta\mu}(f(x_0) - f^* + \mu \|z_0 - x^*\|^2). \tag{31}$$

For $0 \leq s \leq t$, since $\|\nabla f(x_s)\| \leq G$, then Lemma 3.5 implies

$$\|\nabla f(x_s)\|^2 \leq 2L(f(x_s) - f^*). \tag{32}$$

Note that by Algorithm 2, we have

$$z_t - x_t = (1 - \eta\mu\delta_{t-1})(1 - \tau_{t-1})(z_{t-1} - x_{t-1}) + \eta(1 - \delta_{t-1})\nabla f(y_{t-1}).$$

Thus

$$z_t - x_t = \eta \sum_{s=0}^{t-1}(1 - \delta_s)\nabla f(y_s) \prod_{i=s+1}^{t-1}(1 - \eta\mu\delta_i)(1 - \tau_i).$$

Therefore

$$y_t - x_t = \eta\tau_t \sum_{s=0}^{t-1}(1 - \delta_s)\nabla f(y_s) \prod_{i=s+1}^{t-1}(1 - \eta\mu\delta_i)(1 - \tau_i).$$

Moreover

$$1 - \eta\mu\delta_i = 1 - \frac{\eta\mu(A_{i+1} - A_i)}{1 + \eta\mu A_{i+1}} = \frac{1 + \eta\mu A_i}{1 + \eta\mu A_{i+1}}$$

and

$$1 - \tau_i = 1 - \frac{(A_{i+1} - A_i)(1 + \eta\mu A_i)}{A_{i+1} + 2\eta\mu A_i A_{i+1} - \eta\mu A_i^2} = \frac{A_i(1 + \eta\mu A_{i+1})}{A_{i+1} + 2\eta\mu A_i A_{i+1} - \eta\mu A_i^2} \leq \frac{A_i(1 + \eta\mu A_{i+1})}{A_{i+1}(1 + \eta\mu A_i)}.$$

Thus we have

$$\|y_t - x_t\| \leq \eta\tau_t \sum_{s=0}^{t-1}(\delta_s - 1)\frac{A_{s+1}}{A_t} \|\nabla f(y_s)\| \leq \eta \sum_{s=0}^{t-1}\frac{A_{s+1}}{A_t} \|\nabla f(y_s)\| =: \mathcal{I},$$

where the second inequality follows from Lemma D.4. We further control term $\mathcal{I}$ by

$$\mathcal{I} \leq \eta \sum_{s=0}^{t-1}\frac{A_{s+1}}{A_t}(\|\nabla f(x_{s+1})\| + \eta L \|\nabla f(y_s)\|)$$

$$\leq \eta L\mathcal{I} + \eta \sum_{s=0}^{t-1}\frac{A_{s+1}}{A_t} \|\nabla f(x_{s+1})\|.$$

Thus we have

$$\|y_t - x_t\| \leq \frac{\eta}{1 - \eta L} \sum_{s=0}^{t-1} \frac{A_{s+1}}{A_t} \|\nabla f(x_{s+1})\|$$

$$\leq \frac{\eta}{1 - \eta L} \sum_{s=0}^{t-1} \frac{A_{s+1}}{A_t} \sqrt{2L(f(x_{s+1}) - f^*)} \qquad \text{(by (32))}$$

$$\leq \frac{\eta}{1 - \eta L} \sum_{s=0}^{t-1} \frac{A_{s+1}}{A_t} \sqrt{2L \cdot \frac{1}{A_{s+1}\eta\mu}(f(x_0) - f^* + \mu \|z_0 - x^*\|^2)} \qquad \text{(by (31))}$$

$$= \frac{\sqrt{2\eta L(f(x_0) - f^* + \mu \|z_0 - x^*\|^2)}}{(1 - \eta L)\sqrt{\mu}} \sum_{s=0}^{t-1} \frac{\sqrt{A_{s+1}}}{A_t}$$

$$\leq \frac{\sqrt{2\eta L(f(x_0) - f^* + \mu \|z_0 - x^*\|^2)}}{(1 - \eta L)\sqrt{\mu}} \left(3 + 4\log(e + \frac{1}{\eta\mu})\right). \qquad \text{(by Lemma D.7)}$$

$$\leq \frac{\sqrt{\eta}}{1 - \eta L} \left(3 + 4\log(e + \frac{1}{\eta\mu})\right) \cdot \frac{G \cdot L^{1/2 - 1/\alpha}}{4} \qquad \text{(by (27))}$$

$$\leq \frac{3 + 4\log(e + \frac{1}{\eta\mu})}{\log^2\left(e + \frac{144 L^{3 - 2/\alpha}}{\mu}\right)} \cdot \frac{G}{24L} \qquad \text{(by (28))}$$

$$\leq \frac{G}{2L} \leq r(G).$$

Since $\|\nabla f(x_t)\| \leq G$ and we just showed $\|x_t - y_t\| \leq r(G)$, by Lemma 3.3, we have

$$\|\nabla f(y_t)\| \leq \|\nabla f(x_t)\| + L \|y_t - x_t\|$$

$$\leq \sqrt{\frac{2L}{\eta\mu A_t}((f(x_0) - f^*) + \mu \|z_0 - x^*\|^2)} + L \cdot \frac{G}{2L} \qquad \text{(by (31))}$$

$$\leq G\left(\frac{1}{4} + \frac{1}{2}\right) \leq G. \qquad \text{(by } A_t \geq 1/(\eta\mu) \text{ and (27))}$$

Then we complete the induction as well as the proof. $\qquad \square$

*Proof of Theorem D.1.* Combining Lemmas D.8, D.9, and D.10, we know the following inequality holds for all $t \geq 0$.

$$A_{t+1}(f(x_{t+1}) - f^*) + \frac{1 + \eta\mu A_{t+1}}{2\eta} \|z_{t+1} - x^*\|^2 \leq A_t(f(x_t) - f^*) + \frac{1 + \eta\mu A_t}{2\eta} \|z_t - x^*\|^2.$$

Then by telescoping, we get

$$A_t(f(x_t) - f^*) + \frac{1 + \eta\mu A_t}{2\eta} \|z_t - x^*\|^2 \leq A_0(f(x_0) - f^*) + \frac{1 + \eta\mu A_0}{2\eta} \|z_0 - x^*\|^2.$$

Finally, applying Lemma D.5, we have $A_t = B_t + 1/(\eta\mu) \geq 1/(1 - \sqrt{\eta\mu})^{t-1} + 1/(\eta\mu)$. Thus completes the proof. $\qquad \square$

# E  Analysis of GD for non-convex functions

In this section, we provide the proofs related to analysis of gradient descent for non-convex function, including those of Lemma 5.1 and Theorem 5.2.

*Proof of Lemma 5.1.* First, based on Corollary 3.6, we know $\|\nabla f(x)\| \leq G < \infty$. Also note that

$$\|x^+ - x\| = \|\eta \nabla f(x)\| \leq \eta G \leq G/L.$$

Then by Lemma 3.3 and Remark 3.4, we have $x^+ \in \mathcal{X}$ and

$$
\begin{aligned}
f(x^+) &\leq f(x) + \langle \nabla f(x_t), x^+ - x \rangle + \frac{L}{2} \left\| x^+ - x \right\|^2 \\
&= f(x) - \eta(1 - \eta L/2) \left\| \nabla f(x) \right\|^2 \\
&\leq f(x).
\end{aligned}
$$

$\square$

*Proof of Theorem 5.2.* By Lemma 5.1, using induction, we directly obtain $f(x_t) \leq f(x_0)$ for all $t \geq 0$. Then by Corollary 3.6, we have $\|\nabla f(x_t)\| \leq G$ for all $t \geq 0$. Following the proof of Lemma 5.1, we can similarly show

$$
f(x_{t+1}) - f(x_t) \leq \eta(1 - \eta L/2) - \frac{\eta}{2} \left\| \nabla f(x_t) \right\|^2 \leq -\frac{\eta}{2} \left\| \nabla f(x_t) \right\|^2.
$$

Taking a summation over $t < T$ and rearranging terms, we have

$$
\frac{1}{T} \sum_{t<T} \left\| \nabla f(x_t) \right\|^2 \leq \frac{2(f(x_0) - f(x_T))}{\eta T} \leq \frac{2(f(x_0) - f^*)}{\eta T}.
$$

$\square$

## F  Analysis of SGD for non-convex functions

In this section, we provide the detailed convergence analysis of stochastic gradient descent for $\ell$-smooth and non-convex functions where $\ell$ is sub-quadratic.

We first present some useful inequalities related to the parameter choices in Theorem 5.3.

**Lemma F.1.** *Under the parameters choices in Theorem 5.3, the following inequalities hold.*

$$
\eta G \sqrt{2T} \leq 1/2, \quad \eta^2 \sigma L T \leq 1/2, \quad 100 \eta^2 T \sigma^2 L^2 \leq \delta G^2.
$$

*Proof of Lemma F.1.* First note that by Corollary 3.6, we know

$$
G^2 = 2LF = 16L(f(x_0) - f^* + \sigma)/\delta \geq 16L\sigma/\delta,
$$

i.e., $\sigma L \leq G^2 \delta / 16$. Then since we choose $\eta \leq \frac{1}{4G\sqrt{T}}$, we have

$$
\begin{aligned}
\eta G \sqrt{2T} &\leq \sqrt{2}/4 \leq 1/2, \\
\eta^2 \sigma L T &\leq \eta^2 T G^2 \delta / 16 \leq \delta/256 \leq 1/2, \\
100 \eta^2 T \sigma^2 L^2 &\leq 100 \eta^2 T G^4 \delta^2 / 256 \leq \delta G^2.
\end{aligned}
$$

$\square$

Next, we show the useful lemma which bounds $\mathbb{E}[f(x_\tau) - f^*]$ and $\mathbb{E}\left[ \sum_{t<\tau} \|\nabla f(x_t)\|^2 \right]$ simultaneously.

**Lemma F.2.** *Under the parameters choices in Theorem 5.3, the following inequality holds*

$$
\mathbb{E}\left[ f(x_\tau) - f^* + \frac{\eta}{2} \sum_{t<\tau} \|\nabla f(x_t)\|^2 \right] \leq f(x_0) - f^* + \sigma.
$$

*Proof of Lemma F.2.* If $t < \tau$, by the definition of $\tau$, we know $f(x_t) - f^* \leq F$ and $\|\epsilon_t\| \leq \frac{G}{5\eta L}$, and the former also implies $\|\nabla f(x_t)\| \leq G$ by Corollary 3.6. Then we can bound

$$
\|x_{t+1} - x_t\| = \eta \|g_t\| \leq \eta(\|\nabla f(x_t)\| + \|\epsilon_t\|) \leq \eta G + \frac{G}{5L} \leq \frac{G}{L},
$$

where we use the choice of $\eta \leq \frac{1}{2L}$. Then based on Lemma 3.3 and Remark 3.4, for any $t < \tau$, we have

$$
\begin{aligned}
f(x_{t+1}) - f(x_t) \leq & \langle \nabla f(x_t), x_{t+1} - x_t \rangle + \frac{L}{2} \|x_{t+1} - x_t\|^2 \\
= & -\eta \langle \nabla f(x_t), g_t \rangle + \frac{\eta^2 L}{2} \|g_t\|^2 \\
\leq & -\eta \|\nabla f(x_t)\|^2 - \eta \langle \nabla f(x_t), \epsilon_t \rangle + \eta^2 L \|\nabla f(x_t)\|^2 + \eta^2 L \|\epsilon_t\|^2 \\
\leq & -\frac{\eta}{2} \|\nabla f(x_t)\|^2 - \eta \langle \nabla f(x_t), \epsilon_t \rangle + \eta^2 L \|\epsilon_t\|^2,
\end{aligned}
\tag{33}
$$

where the equality is due to (4); the second inequality uses $g_t = \epsilon_t + \nabla f(x_t)$ and Young's inequality $\|y + z\|^2 \leq 2 \|y\|^2 + 2 \|z\|^2$ for any vectors $y, z$; and the last inequality chooses $\eta \leq 1/(2L)$. Taking a summation over $t < \tau$ and rearanging terms, we have

$$
f(x_\tau) - f^* + \frac{\eta}{2} \sum_{t < \tau} \|\nabla f(x_t)\|^2 \leq f(x_0) - f^* - \eta \sum_{t < \tau} \langle \nabla f(x_t), \epsilon_t \rangle + \eta^2 L \sum_{t < \tau} \|\epsilon_t\|^2 .
$$

Now we bound the last two terms on th RHS. First, for the last term, we have

$$
\mathbb{E} \left[ \sum_{t < \tau} \|\epsilon_t\|^2 \right] \leq \mathbb{E} \left[ \sum_{t < T} \|\epsilon_t\|^2 \right] \leq \sigma^2 T,
$$

where the first inequality uses $\tau \leq T$ by its defnition; and in the last inequality we use Assumption 4.

For the cross term, note that $\mathbb{E}_{t-1} \left[ \langle \nabla f(x_t), \epsilon_t \rangle \right] = 0$ by Assumption 4. So this term is a sum of a martingale difference sequence. Since $\tau$ is a stopping time, we can apply the optional stopping theorem to obtain

$$
\mathbb{E} \left[ \sum_{t \leq \tau} \langle \nabla f(x_t), \epsilon_t \rangle \right] = 0.
\tag{34}
$$

Then we have

$$
\mathbb{E} \left[ -\sum_{t < \tau} \langle \nabla f(x_t), \epsilon_t \rangle \right] = \mathbb{E} \left[ \langle \nabla f(x_\tau), \epsilon_\tau \rangle \right] \leq G \mathbb{E}[\|\epsilon_\tau\|] \leq G \sqrt{\mathbb{E}[\|\epsilon_\tau\|^2]}
$$

$$
\leq G \sqrt{\mathbb{E} \left[ \sum_{t \leq T} \|\epsilon_t\|^2 \right]} \leq \sigma G \sqrt{T + 1} \leq \sigma G \sqrt{2T},
$$

where the equality is due to (34); the first inequality uses $\|\nabla f(x_\tau)\| \leq G$ by the definition of $\tau$ in (5) and Corollary 3.6; the fourth inequality uses $\mathbb{E}[X]^2 \leq \mathbb{E}[X^2]$ for any random variable $X$; and the last inequality uses Assumption 4.

Combining all the bounds above, we get

$$
\mathbb{E} \left[ f(x_\tau) - f^* + \frac{\eta}{2} \sum_{t < \tau} \|\nabla f(x_t)\|^2 \right] \leq f(x_0) - f^* + \eta \sigma G \sqrt{2T} + \eta^2 \sigma^2 L T
$$

$$
\leq f(x_0) - f^* + \sigma,
$$

where the last inequality is due to Lemma F.1. $\qquad \square$

With Lemma F.2, we are ready to prove Theorem 5.3.

*Proof of Theorem 5.3.* We want to show the probability of $\{\tau < T\}$ is small, as its complement $\{\tau = T\}$ means $f(x_t) - f^* \leq F$ for all $t \leq T$ which implies $\|\nabla f(x_t)\| \leq G$ for all $t \leq T$. Note that

$$
\{\tau < T\} = \{\tau_2 < T\} \cup \{\tau_1 < T, \tau_2 = T\}.
$$

Therefore we only need to bound the probability of each of these two events on the RHS.

We first bound $\mathbb{P}(\tau_2 < T)$. Note that

$$
\begin{aligned}
\mathbb{P}(\tau_2 < T) =& \mathbb{P}\left(\bigcup_{t<T}\left\{\|\epsilon_t\| > \frac{G}{5\eta L}\right\}\right) \\
\leq & \sum_{t<T}\mathbb{P}\left(\|\epsilon_t\| > \frac{G}{5\eta L}\right) \\
\leq & \frac{25\eta^2 T\sigma^2 L^2}{G^2} \\
\leq & \delta/4,
\end{aligned}
$$

where the first inequality uses union bound; the second inequality applies Chebyshev's inequality and $\mathbb{E}[\|\epsilon_t\|^2] = \mathbb{E}[\mathbb{E}_{t-1}[\|\epsilon_t\|^2]] \leq \sigma^2$ for each fixed $t$ by Assumption 4; the last inequality uses Lemma F.1.

Next, we will bound $\mathbb{P}(\tau_1 < T, \tau_2 = T)$. Note that under the event $\{\tau_1 < T, \tau_2 = T\}$, we know that 1) $\tau = \tau_1 < T$ which implies $f(x_{\tau+1}) - f^* > F$; and 2) $\tau < T = \tau_2$ which implies $\|\epsilon_\tau\| \leq \frac{G}{5\eta L}$ by the definition in (5). Also note that we always have $f(x_\tau) - f^* \leq F$ which implies $\|\nabla f(x_\tau)\| \leq G$ by Corollary 3.6. Then we can show

$$
\|x_{\tau+1} - x_\tau\| = \eta\|g_\tau\| \leq \eta(\|\nabla f(x_\tau)\| + \|\epsilon_\tau\|) \leq \eta G + \frac{G}{5L} \leq \frac{G}{L},
$$

where we choose $\eta \leq \frac{1}{2L}$. Then based on Lemma 3.3 and Remark 3.4, we have

$$
\begin{aligned}
f(x_{\tau+1}) - f(x_\tau) \leq & -\frac{\eta}{2}\|\nabla f(x_\tau)\|^2 - \eta\langle\nabla f(x_\tau), \epsilon_\tau\rangle + \eta^2 L\|\epsilon_\tau\|^2 \\
\leq & \eta\|\nabla f(x_\tau)\| \cdot \|\epsilon_\tau\| + \eta^2 L\|\epsilon_\tau\|^2 \\
\leq & \frac{G^2}{4L} \\
= & \frac{F}{2},
\end{aligned}
$$

where the first inequality is obtained following the same derivation as in (33); the last equality is due to Corollary 3.6. Therefore we can show that under the event $\{\tau_1 < T, \tau_2 = T\}$,

$$
f(x_\tau) - f^* = f(x_\tau) - f(x_{\tau+1}) + f(x_{\tau+1}) - f^* > F/2.
$$

Hence,

$$
\mathbb{P}(\tau_1 < T, \tau_2 = T) \leq \mathbb{P}\left(f(x_\tau) - f^* > F/2\right) \leq \frac{\mathbb{E}[f(x_\tau) - f^*]}{F/2} \leq \frac{2(f(x_0) - f^* + \sigma)}{F} = \delta/4,
$$

where the second inequality uses Markov's inequality; the third inequality uses Lemma F.2; and in the last inequality we choose $F = 8(f(x_0) - f^* + \sigma)/\delta$.

Therefore we can show

$$
\mathbb{P}(\tau < T) \leq \mathbb{P}(\tau_2 < T) + \mathbb{P}(\tau_1 < T, \tau_2 = T) \leq \delta/2.
$$

Then we also know $\mathbb{P}(\tau = T) \geq 1 - \delta/2 \geq 1/2$. Therefore, by Lemma F.2,

$$
\begin{aligned}
\frac{2(f(x_0) - f^* + \sigma)}{\eta} \geq & \mathbb{E}\left[\sum_{t<\tau}\|\nabla f(x_t)\|^2\right] \\
\geq & \mathbb{P}(\tau = T)\mathbb{E}\left[\sum_{t<T}\|\nabla f(x_t)\|^2 \,\middle|\, \tau = T\right] \\
\geq & \frac{1}{2}\mathbb{E}\left[\sum_{t<T}\|\nabla f(x_t)\|^2 \,\middle|\, \tau = T\right].
\end{aligned}
$$

Then we have

$$\mathbb{E}\left[\frac{1}{T}\sum_{t<T}\|\nabla f(x_t)\|^2 \,\middle|\, \tau = T\right] \le \frac{4(f(x_0) - f^* + \sigma)}{\eta T} = \frac{\delta F}{2\eta T} \le \frac{\delta}{2}\cdot\epsilon^2,$$

where the last inequality uses the choice of $T$. Let $\mathcal{E} := \{\frac{1}{T}\sum_{t<T}\|\nabla f(x_t)\|^2 > \epsilon^2\}$ denote the event of not converging to an $\epsilon$-stationary point. By Markov's inequality, we have $\mathbb{P}(\mathcal{E}) \le \delta/2$. Therefore we have $\mathbb{P}(\{\tau < T\}\cup\mathcal{E}) \le \delta$, which completes the proof. $\qquad\square$

## G Lower bound

In this section, we provide the proof of Theorem 5.4.

*Proof of Theorem 5.4.* Let $c, \eta_0 > 0$ satisfy $\eta_0 \le c^2/2$. Consider

$$f(x) = \begin{cases} \log(|x| - c), & |x| \ge y \\ 2\log(y - c) - \log(2y - |x| - c), & c/2 \le |x| < y \\ kx^2 + b, & |x| < c/2, \end{cases}$$

where $c > 0$ is a constant and $y = (c + \sqrt{c^2 + 2\eta_0})/2 > 0$ is the fixed point of the iteration

$$x_{t+1} = \left|x_t - \frac{\eta_0}{x_t - c}\right|,$$

and $k$, $b$ are chosen in such a way that $f(x)$ and $f'(x)$ are continuous. Specifically, choose $k = c^{-1}f'(c/2)$ and $b = f(c/2) - cf'(c/2)/4$. Since $f(-x) = f(x)$, $f(x)$ is symmetric about the line $x = 0$. In a small neighborhood, $f(x)$ is symmetric about $(y, f(y))$, so $f'(x)$ is continuous at $y$.

Let us first consider the smoothness of $f$. By symmetry, it suffices to consider $x > 0$. Then,

$$f'(x) = \begin{cases} (x - c)^{-1}, & x \ge y \\ (2y - x - c)^{-1}, & c/2 \le x < y \\ 2kx, & 0 < x < c/2. \end{cases}$$

Its Hessian is given by

$$f''(x) = \begin{cases} -(x - c)^{-2}, & x > y \\ (2y - x - c)^{-2}, & c/2 < x < y \\ 2k, & 0 < x < c/2. \end{cases}$$

Hence, $f(x)$ is $(2, 2k, 1)$-smooth.

Note that $f(x)$ has a stationary point $0$. For stepsize $\eta_f$ satisfying $\eta_0 \le \eta_f \le c^2/4$, there exists $z = (c + \sqrt{c^2 + 2\eta_f}) \ge y$ such that $-z = z - \eta_f(y - c)^{-1}$ and by symmetry, once $x_\tau = z$, $x_t = \pm z$ for all $t \ge \tau$, making the GD iterations stuck. Now we choose a proper $x_0$ such that $f'(x_0)$ and $f(x_0) - f(0)$ are bounded.

We consider arriving at $y$ from above. That is, $x_0 \ge x_1 \ge \ldots x_\tau = z > c > 0$. Since in each update where $x_{t+1} = x_t - \eta_f(x_t - c)^{-1} > c$,

$$x_t - x_{t+1} = x_t - (x_t - \eta_f(x_t - c)^{-1}) = \eta_f(x_t - c)^{-1} \le \sqrt{\eta_f}.$$

Hence, we can choose $\tau$ in such a way that $3c/2 \le x_0 < 3c/2 + \sqrt{\eta_f}$. Then,

$$\log(c/2) \le f(x_0) \le \log(c/2 + \sqrt{\eta_f}), \quad 2/(c + 2\sqrt{\eta_f}) \le f'(x_0) \le 2/c.$$

By definition, $y - c = \eta_0(c + \sqrt{c^2 + 2\eta_0})^{-1}$. Hence,

$$\begin{aligned} f(c/2) &= 2\log(y - c) - \log(2y - c/2 - c) \\ &= 2\log(\eta_0) - 2\log(c + \sqrt{c^2 + 2\eta_0}) - \log(\sqrt{c^2 + 2\eta_0} - c/2), \\ f'(c/2) &= \frac{1}{\sqrt{c^2 + 2\eta_0} - c/2} \end{aligned}$$

Then,

$$
\begin{aligned}
f(x_0) - f(0) &= f(x_0) - f(c/2) + cf'(c/2)/4 \\
&\leq \log(c/2 + \sqrt{\eta_f}) + 2\log(\eta_0^{-1}) + 2\log(c + \sqrt{c^2 + 2\eta_0}) \\
&\quad + \log(\sqrt{c^2 + 2\eta_0} - c/2) + \frac{c}{4}\frac{1}{\sqrt{c^2 + 2\eta_0} - c/2} \\
&\leq \log(c) + 2\log(\eta_0^{-1}) + 2\log(2\sqrt{2c^2}) + \log(\sqrt{2c^2}) + \frac{1}{2} \\
&= 4\log(c) + 2\log(\eta_0^{-1}) + \frac{7}{2}\log(2) + \frac{1}{2}.
\end{aligned}
$$

For stepsize $\eta_f < \eta_0$, reaching below $4c/3$ takes at least

$$
(x_0 - 4c/3)/\sqrt{\eta_f} \geq c/(6\sqrt{\eta_f}) > c\eta_0^{-1/2}/6
$$

steps to reach $4c/3$, where $f'(4c/3) = \log(c/3)$.

Now we set $c$ and $\eta_0$ and scale function $f(x)$ to satisfy the parameter specifications $L_0, L_2, G_0, \Delta_0$. Define $g(x) = L_2^{-1}f(x)$. Then, $g(x)$ is $(2, 2kL_2^{-1}, L_2)$-smooth. Since the gradient of $g(x)$ is $L_2^{-1}$ times $f(x)$, the above analysis for $f(x)$ applies to $g(x)$ by replacing $\eta_0$ with $\eta_1 = L_2\eta_0$ and $\eta_f$ with $\eta = L_2\eta_f$. To ensure that

$$
2kL_2^{-1} = 2(cL_2)^{-1}f'(c/2) = \frac{2}{cL_2}\frac{1}{\sqrt{c^2 + 2\eta_1} - c/2} \leq \frac{4}{c^2 L_2} \leq L_0,
$$

it suffices to take $c \geq 2/\sqrt{L_0 L_2}$. To ensure that

$$
g'(x_0) \leq \frac{2}{L_2 c} \leq G_0,
$$

it suffices to take $c \geq 2/(L_2 G_0)$. To ensure that

$$
g(x_0) - g(0) \leq (4\log(c) + 2\log(\eta_1^{-1}) + 3.5\log 2 + 0.5)L_2^{-1} \leq \Delta_0,
$$

it suffices to take

$$
\log(\eta_1^{-1}) = \frac{L_2\Delta_0 - 3.5\log 2 - 0.5}{2} - 2\log(c).
$$

Since we require $\eta_1 \leq c^2/2$, parameters $L_2$ and $\Delta_0$ need to satisfy

$$
\log 2 - 2\log(c) \leq \frac{L_2\Delta_0 - 3.5\log 2 - 0.5}{2} - 2\log(c),
$$

that is, $L_2\Delta_0 \geq 5.5\log 2 + 0.5$, which holds because $L_2\Delta_0 \geq 10$. Take $c = \max\{2/\sqrt{L_0 L_2}, 2/(L_2 G_0), \sqrt{8/L_0}\}$. Then, as long as $\eta \leq 2/L_0$, the requirement that $\eta \leq c^2/4$ is satisfied. Therefore, on $g(x)$ with initial point $x_0$, gradient descent with a constant stepsize either gets stuck, or takes at least

$$
\begin{aligned}
c\eta_1^{-1/2}/6 &= \frac{c}{6}\exp\left(\frac{L_2\Delta_0 - 3.5\log 2 - 0.5}{4} - \log(c)\right) \\
&= \frac{1}{6}\exp(\frac{L_2\Delta_0 - 3.5\log 2 - 0.5}{4}) \\
&\geq \frac{1}{6}\exp(\frac{L_2\Delta_0}{8})
\end{aligned}
$$

steps to reach a 1-stationary point.

On the other hand, if $\eta > 2/L_0$, consider the function $f(x) = \frac{L_0}{2}x^2$. For any $x_t \neq 0$, we always have $|x_{t+1}|/|x_t| = |1 - \eta L_0| > 1$, which means the iterates diverge to infinity. $\square$

