# OpenReview forum: "Convex and Non-convex Optimization Under Generalized Smoothness"
_NeurIPS.cc/2023/Conference — NeurIPS 2023 spotlight_

### Official Review · Reviewer_YkEg · 2023-06-26

**Soundness:** 2 fair
**Presentation:** 2 fair
**Contribution:** 1 poor
**Rating:** 3
**Confidence:** 4

**Summary:**

This submission claims to relax the Lipschitz assumption on the gradient of the objective function in nonconvex optimisation, and obtains convergence bounds similar to textbook convergence results.

**Strengths:**

The presentation of the material is clear and the narrative flows well.

**Weaknesses:**

The result presented in this manuscript is not of importance. When we take the gradient information about the initialisation point into consideration, we can correspondingly restrict the feasibility set to the neighbourhood around the initialisation point, and then the $(r,l)$-smoothness defined in this paper can be replaced by the classical Lipschitz condition.

**Questions:**

In equation (5), why is there a $-1$ in the definition of $S_{\textrm{rect}}$?

**Limitations:**

As there is no novel discovery in this submission, the limitation discussion is not applicable.

---

> ### Author Rebuttal · Authors · 2023-08-08
>
> We would like to thank the Reviewer for the comments. However, we do NOT think there is any simple way to restrict the feasibility set to the neighborhood around the initialization point based on gradient information around the initialization, as claimed by the reviewer.
>
> In particular, we want to clarify the following points regarding the challenges of restricting the feasibility set to some neighobood:
> - First, stationary points may be very far away from the initialization. For example, consider the function $f=\exp(-x)$ with domain $\mathbb{R}$, whose stationary point is at infinity. If you restrict the feasible set within a neighborhood of initialization, you will only converge to a sub-optimal point, which makes no sense.
> - For another example, consider the function $f(x)=1/x+1/(1-x)$ with domain $(0,1)$. Suppose the initialization is close to $0$, it is hard to make sure the neighborhood contains the optimal point at $1/2$ and does not go outside of the domain, given that the initialization point is closer to the boundary than the optimal point. Although it might be possible to find such a neighborhood for this simple convex and one-dimensional example (e.g. by cheating), we believe it is hard for a non-convex and high-dimensional function.
> - We also want to point out that the boundary of the domain could be non-convex and not known to the algorithm, which makes things even harder. So you may end up getting a non-convex neighborhood in order to contain stationary points, which does not really reduce the problem to the classical smoothness condition, since classical analysis requires a convex feasible set.
>
> Based on the discussions above, we respectfully disagree with the reviewers' suggestion that our results are not novel or important.
>
> Regarding the question on equation (5), the $-1$ is there because we do not include the two end points $\tau$ and $\tau_{1/2}$ when defining $S_{\text{rect}}$, in other words, $S_{\text{rect}}:=\sum_{\tau_{1/2}<t<\tau} (G/2)^2=(G/2)^2(\tau-\tau_{1/2}-1)$

---

### Official Review · Reviewer_QDyc · 2023-06-27

**Soundness:** 4 excellent
**Presentation:** 3 good
**Contribution:** 3 good
**Rating:** 7
**Confidence:** 4

**Summary:**

Relaxed smoothness conditions have been introduced to study the gradient clipping algorithm, and show that clipping in particular allows fixed step-size convergence without smoothness under this relaxed assumption. This paper further generalizes the relaxed smoothness notion used for clipping by bounding the Hessian by any non-decreasing function of the gradient norm.

Then, fixed step-size convergence is shown under the relaxed smoothness assumption in a variety of settings. The key technical part is to show that assuming a large enough initial bound on the gradients and relaxed smoothness, the gradients remain bounded throughout the trajectory regardless of the setting (convex, strongly convex, non-convex, and even stochastic with some necessary caveats). Then, boundedness of the gradients along the trajectory implies the regular smoothness condition, and thus standard analyses can be used to show convergence of gradient descent.

**Strengths:**

- Shows convergence of GD, NAG and SGD for relaxed smoothness conditions, which were not known without gradient clipping. This allows to prove convergence of GD for general classes of functions.
- Show boundedness of the gradients along trajectories under relaxed smoothness conditions. This is a nice-to-have technical result, especially for NAG.

**Weaknesses:**

- Fixed step-size rates seem quite pessimistic in this case, since step-sizes all along the trajectory depend on the initial bound on smoothness, which might be very large.

**Questions:**

1) The main argument is that global bounds hold on the gradients. Although it is good to have, this potentially leads to very small step-sizes even in regions for which gradients are small.
Would it be possible to improve those bounds, and show that the gradients actually decrease along the trajectories (e.g., in the strongly convex setting), in order to allow for increasingly large step-sizes? If so, what would the decrease rate be?

2) How do results in the paper compare with simple clipped (S)GD with clipping threshold set to the gradient norm at $x_0$ (basically, enforce boundedness instead of showing it) ? Convergence rates seem equivalent but additional terms seem different in each case. Could you clarify that?




**Limitations:**

The authors have addressed the limitations of their work adequately.

---

> ### Author Rebuttal · Authors · 2023-08-09
>
> We thank the reviewer for the positive comments! We will try to address the concerns and questions below.
>
> 1. Regarding the stepsize, we agree that using an adaptive stepsize (e.g. gradient clipping technique, which essentially uses a larger step size when gradients are small) may accelerate convergence. However, only constant-factor acceleration is possible because the convergence rates in our paper are already optimal up to constant factors. This may matter in practice but is not important for theoretical analysis. Actually, our analysis should directly apply to most methods with an adaptive stepsize, at least in the deterministic setting. We consider using a constant stepsize for the following reasons:
> - There are already a lot of papers studying methods with an adaptive stepsize for $(L_0,L_1)$-smooth functions. We are the first to study the classical methods with a constant stepsize and our results show that adaptivity is not necessary for generalized smooth functions.
> - In the stochastic setting, using a constant step size allows the relaxation of the noise assumption from the assumption of bounded noise in existing papers to that of bounded variance in our paper.
> - For NAG, using adaptive stepsize may lead to a very sophisticated algorithm, which is hard to implement.
>
> 2. We believe that the simple clipped (S)GD with clipping threshold set to the gradient norm at $x_0$ should also converge with the same rate to constant-stepsize (S)GD, up to constant factors. We believe the convergence can be shown using our analysis. In the convex setting, they are actually equivalent since the gradient norm is non-increasing by Lemma 4.1. In the non-convex setting, it is hard and not very important to theoretically compare different constant factors in their convergence speeds, since we have to also obtain very precise complexity lower bounds for them.

---

### Official Review · Reviewer_GKNp · 2023-07-06

**Soundness:** 4 excellent
**Presentation:** 4 excellent
**Contribution:** 4 excellent
**Rating:** 8
**Confidence:** 4

**Summary:**

This paper generalizes the recently introduced (L0,L1)-smoothness which itself extends the L-smoothness which is key in analyzing rates of convergence of optimization algorithms. The authors introduce the concept of $\ell$-smoothness where $\ell$ is a function of the gradient of the function to minimize (e.g.: a polynomial).
The key idea is to show that under the $\ell$-smoothness assumption, the gradient remains bounded along the iterates of a given algorithm.
The authors focus on smooth optimization and tackle both convex and non-convex settings, in which they (roughly) recover the bounds already known for L-smooth functions and show whether they hold or not with generalized smoothness depending on the function $\ell$.

**Strengths:**

The paper is well written, and the presentation is clear. The mathematical statements are rigorous and I did not spot mistakes. The paper builds on existing works on (L0-L1)-smoothness but significantly extends them and covers many important settings (convex/non-convex, etc.). Additionally, I found the reasoning interesting and the proof techniques used depart from the classical ones (especially in the non-convex setting).
Overall, I think that this paper brings a significant contribution to the important question of the convergence of optimization algorithms. My general feeling is very positive.

**Weaknesses:**

On the theoretical side I think that there are some minor bugs (see questions below), but nothing really problematic.
However, while the paper has extensive theoretical results, it does not provide any numerical experiments, which is important for machine learning and optimization papers. It would have been, for example, very informative to see how the theory allows making GD and NAG converging beyond Lipschitz assumption by using the author's step-sizes conditions.
The related work section seems too short to me given that gradient-based optimization is a very active topic of research. For example the authors call many results as "well known" where instead credit could have been given.

**Questions:**

1) Line 106 on the fact that f has to tend to infinity. First it might be good to specify +infinity (instead of infinity) to remove any ambiguity. Then I think that the statement is not true in its current form since R^d is an open set, yet there, the epigraph of f is closed without coercivity assumptions (as the authors discuss after). I think that the discussion is valid for *bounded* open domains. Could you clarify/correct this?

2) Line 157. Following my previous question, the authors motivate the use of a domain X so as to include functions like logarithms and rational functions. However it seems that these functions are not closed (eg: log(x) when x -> 0), so Assumption 2 does not hold. Could you comment on that?

3) Line 76-78. In the paper, and notably at these lines, there is a confusion between NAG which is indeed optimal for convex functions, and Heavy Ball with Friction (HBF). Indeed, NAG while being similar to HBF is not optimal in the strongly convex setting but HBF is. This should be clarified. (l76-78)

4) On Assumption 3 (existence of a minimizer). While being reasonable this makes the studied framework more restrictive than the classical one (where the optimal rate is also valid for functions whose minimizer is at infinity). Therefore the results are not valid for the whole class of convex functions (but a very large subset of it), this should be a little more emphasized while discussing the contributions so as not to mislead the reader in the introduction.

5) It would be better to define rigorously "sub-quadratic", in particular, does it include quadratic functions or is it strictly sub-quadratic?

6) Do the authors have any idea on whether their modified NAG (Alg. 1) might improve NAG even when used on L-smooth functions? (even though NAG is already theoretically optimal). If not, is it worse than the vanilla version or does it behave similarly? Numerical experiments would be insightful here too.

7) Typos: l76: strong convex -> strongly convex
l207: line 6 of the algorithm -> Is it not rather line 4 of the algorithm?

**Limitations:**

Some limitations are discussed, in particular those related to NAG in the non-convex setting. The limitations about existence of a minimizer might be further discussed (see questions).
Societal impact is not really applicable here.

---

> ### Author Rebuttal · Authors · 2023-08-09
>
> We thank the reviewer for the positive feedback! We will add some experimental results to support our theories in the revision. We will also consider adding more related works. Below we try to address your questions.
>
> 1. First, we do mean positive infinity in Line 106, and thank you for pointing it out. We do not quite understand why the argument "since $\mathbb{R}^d$ is an open set, yet there, the epigraph of f is closed without coercivity assumptions" suggests our statement is not true. We believe it is consistent with our statement. Note that $\mathbb{R}^d$ does not have boundary, and thus in this case, we do NOT require $f$ to go to infinity when $x$ goes to infinity. Please let us know if we misunderstood your question.
>
> 2. We want to clarify that, although Assumptions 1 and 2 are necessary for our convergence analysis,  the definitions of $\ell$-smooth or $(r,\ell)$-smooth functions themselves are independent of these two assumptions. So the examples in this section do not need to satisfy these two assumptions. Of course, we are more interested in examples satisfying them. So the logarithmic function here refers to $-\log(x)$ and rational functions refer mostly to those satisfying these two assumptions.
>
> 3. Thank you for the suggestion! Could you elaborate on why NAG is not optimal for strongly convex functions, given that its complexity is $\sqrt{\kappa}\log(1/\epsilon)$?
>
> 4. Our analysis does directly apply to the case where the optimal point does not exist. We make this assumption just to present the convergence result in terms of the distance between the iterate and the optimal point, so that it looks similar to the most classical textbook ones. We will make it clear in the revision.
>
> 5. We mean strictly sub-quadratic functions, which means $\lim_{u\to\infty}\ell(u)/u^2=0$, and will make it clear in the revision.
>
> 6. Our modified version of NAG has the same rate as the classical one for $L$-smooth functions. The modification is minor and for technical convenience. We are not sure whether the modification is necessary for generalized smooth functions. We will add some empirical results based on the suggestions.
>
> 7. Thank you for pointing out the typos, and we will fix them in the revision.

---

### Official Review · Reviewer_cK5L · 2023-07-06

**Soundness:** 3 good
**Presentation:** 2 fair
**Contribution:** 2 fair
**Rating:** 6
**Confidence:** 4

**Summary:**

This paper introduces a new assumption generalizing classical smoothness, and named $\ell$-smoothness, motivates it by giving providing examples of non-smooth functions belonging to this class, and studies classical algorithms under this assumption.

**Strengths:**

- The paper is clear and fairly compared to related work.
- The new class is well defined and motivated, and authors show that we can obtain results under it.

**Weaknesses:**

$\underline{\text{General remarks}}$:

- First, I would like to make a somewhat subjective statement about the class: in my humble opinion, it seems a bit flawed as it does not respect fundamental homogeneity properties. Let me explain:

Let $f\in\mathcal{F}_{\ell}$ the class of $\ell$-smooth functions. For $\alpha, \lambda > 0$, define $g$ as $g(x)=\frac{\alpha}{\lambda^2}f(\lambda x)$ (assume wlog that their minimum is in 0, otherwise translate $f$, then create $g$).
We verify $\nabla g(x)=\frac{\alpha}{\lambda}\nabla f(\lambda x)$ and $\nabla^2 g(x)=\alpha\nabla^2 f(\lambda x)$,
hence $\|\nabla^2 g(x)\| = \|\alpha\nabla^2 f(\lambda x)\| \leq \alpha \ell( \|\nabla f(\lambda x)\|) = \alpha \ell( \frac{\lambda}{\alpha}\|\nabla g(x)\|)$.

Finally, $g\in\mathcal{F}_{\alpha \ell(\frac{\lambda}{\alpha} .)}$.

Let us assume that after some analysis of an algorithm like GD (this reasoning also applies to momentum), we find out that the step-size that achieves the best worst-case guarantee on the class $\mathcal{F}_{\ell}$ is $\gamma_{\ell}$.

Then, if I need to optimize $f$, I will use $x_{t+1} = x_t - \gamma_{\ell} \nabla f (x_t)$.

Now, instead I minimize $g$, I will use $x_{t+1} = x_t - \gamma_{\alpha \ell(\frac{\lambda}{\alpha} .)} \nabla g (x_t) = x_t - \gamma_{\alpha \ell(\frac{\lambda}{\alpha} .)} \frac{\alpha}{\lambda}\nabla f(\lambda x_t)$.

Note that $f$ and $g$ have the same minimum and if we introduce the iterates $y_t = \lambda x_t$, we have

$x_{t+1} = x_t - \gamma_{\ell} \nabla f (x_t)$ when minimizing $f$,

and

$y_{t+1} = y_t - \gamma_{\alpha \ell(\frac{\lambda}{\alpha} .)} \alpha\nabla f(y_t)$ when minimizing $g$.

In short, we need to have $\gamma_{\ell} = \alpha \gamma_{\alpha \ell(\frac{\lambda}{\alpha} .)}$, or again $ \gamma_{\alpha \ell(\frac{\lambda}{\alpha} .)} = \frac{\gamma_{\ell}}{\alpha}$.
Indeed, they both optimize their dynamics.

This shows that $\lambda$ has no impact on the optimal way to tune an algorithm. And we can stretch the function $\ell$ as much as we want and notice that probably only $\ell(0)$ matters, i.e. the smoothness constant in the optimum.
In particular, applied to the case where $\ell(x) = L_0 + L_1x$, it is clear that the optimal parameter cannot depends on $L_1$.

This observation is explained by the lack of homogeneity in the formula: when scaling a function, only $L_0$ is scaled, not $L_1.$
In $L$-smooth class, the worst-case function generally do not belong to the $L-\varepsilon$-smooth class. Hence there is a hierarchy of the classes, but here, it seems that by scaling a function, the worst-case dynamics does not depend on some part of the class definition, leading to useless specification.

This point can be further discussed. This is just some thoughts I had based on the class definition which I never used and I am aware from the related works section that some people are working with it. Thus I would be happy if authors could discuss this point.

However, based on this remark, my first guess was that the analysis would basically lead as the same analysis as when $\ell$ is constant, or $L_1=0$, which leads to my next point.

- Second, we indeed recover classical analyses everywhere in this paper. There indeed is an additional argument, that is the sequence of iterate stays in a compact and by regularity of $f$, we can bound the gradients, hence the hessians and conclude with all the classical analyses.
Indeed:
    - Th4.2 and 4.3 use classical Lyap analyses.
    - Lemma 4.1: Using cocoercivity, authors prove $\|\nabla f(x_t)\|$ is decreasing.  Using the same proof, one could have that $\|x_t-x_\star\|$ is decreasing as well. And using descent lemma, we have that $f(x_t)$ is also decreasing.
Finally, we can prove the descent lemma without assumption of $\ell$-smoothness by just assuming the continuity of the hessian, and taking G as upper bound of $\lbrace \|\nabla^2 f(x)\| | x\in\mathcal{X} \text{ and } f(x)\leq f(x_0) \rbrace$ which is assumed to be a compact since authors assume that the function tends to infinity on the border of $\mathcal{X}$. We have the revisited descent lemma: $f(x_{t+1}) \leq f(x_t) - \eta\|\nabla f(x_t)\|^2 + \frac{1}{2}\eta^2 G \|\nabla f(x_t)\|^2$, and by taking $\eta$ sufficiently small, we insure in 1 calculus that
    - 1) $f(x_t)$ is decreasing and that all the hessians keep being smaller than $G$.
    - 2) the squared gradients are sommable, hence the classical complexity $O(1/\varepsilon^2)$ that is as in thm5.1, obtained in a much simpler way.  Of course, here we only assumed the hessian to be bounded, so $\ell$-smoothness + bounded gradients do the job.

In conclusion, in my opinion, most of the results are almost straightforward from what is known in the literature.

- Third, in my opinion, the stochastic assumption A4 is too strong. I am aware authors claim that some works in the literature assume even stronger assumption, but A4 is way too strong: no multiplicative noise, only additive. This paper claims generalizing smoothness, but linear regressions with MSE losses are not even covered by the section 5.2.  Plus, bounded variance are too easy to handle in general and leads to an analysis close the deterministic case.  Instead, authors should consider expected smoothness (or its equivalent in $\ell$-smoothness). Actually, under the assumption that there is a finite number of functions under consideration, I would guess we could generalize the same arguments as in the deterministic case to ensure all the hessians to be bounded on the optimization iterates and basically use classical SGD proof in the smooth case.

$~$

$\underline{\text{Minor}}$:

- Clarity:

    - Assumption 1: First recall definition of « closed function »
    - Also define « sub-quadratic »
    - l.468: It took me a while to find where this proof was. Please do as for other propositions: state it right before the proof. In a general way, please state all theorems right before their proofs if reported in appendix. Use the « restatable » latex package too avoid renumbering.
    - Proof of lemma B1: From convexity and local smoothness, authors apply the $\underline{\text{descent lemma}}$ on the $\underline{\text{Bregman divergence}}$ of the objective function to obtain local $\underline{\text{cocoercivity}}$. The proof is the same as for global smoothness. Yet, for completness and care of the neighborhood needed to obtain the cocoercivity, I understand that the proof is provided. However, It needs a reference to this classical result in the global smooth case and mention of the 3 underlined terms when used.
        - l.550: Cocoercivity
        - l.553: Bregman divergences
        - l.562: Descent Lemma
    - l.560: please introduce y: « let y [as in the lemma statement] »
    - l.248: « Theorem 5.1 gives the classical $O(1/T)$ rate, or $O(1/\varepsilon^2)$ gradient complexity » -> $O(1/\sqrt{T})$. Authors do not clearly state if they are talking here about gradient norm or its square, but they need to be consistent when talking about rate and complexity.

- Typos:
    - l.2: "Lipshitzness" -> Lipschitz continuity.
    - l.76: "strong convex" -> strongly convex
    - l.79: $\nabla$ is missing
    - l.144: « $x = x_2 = x_t$ » ? I guess "$x$" needs to be removed
    - l.169: « accelearted » -> accelerated
    - l.570: 1/L -> 2/L


- Misc:
    - Table 1: missing hline + be precise on the meaning of « - ».
    - Table 1: for GD non convex: « Inf or $\Omega$ … » could be summed up as «  $\Omega$ … ».
    - l.121-122: Put 2) under 1). It should not exceed the current number of lines.

**Questions:**

Can authors discuss my first point?

Also, can they try to study SGD under expected smoothness assumption?

---

> ### Author Rebuttal · Authors · 2023-08-08
>
> We would to thank the reviewer for the insightful thoughts and comments! Below we will clarify the three points in the review.
>
> **1. Regarding the first point**, the reviewer made a very interesting reasoning regarding the optimal worse-case step-size $\gamma_{\ell}$, as defined in the comment. However, we want to point out that, for our $\ell$-smooth functions, **the stepsize has to depend on the initialization point**, in addition to $\ell$. Otherwise, let us consider the simple convex function $1/x+1/(1-x)$ with domain $(0,1)$. For any positive stepsize independent of the initial point, we can always find some initialization point $x_0$ close enough to $0$, whose gradient is large enough so that after one step of gradient descent, $x_1>1$ goes outside of the domain.
>
> Let the optimal worse-case stepsize be $\gamma_{\ell, \\|\nabla f(x_0)\\|}$ which depends on both $\ell$ and $\\|\nabla f(x_0)\\|$, as in all our theorems. Following your reasoning, we can obtain something like $\gamma_{\ell, \\|\nabla f(x_0)\\|}=\alpha \gamma_{\alpha \ell(\frac{\lambda}{\alpha}\cdot), \\|\nabla g(x_0/\lambda)\\|}=\alpha \gamma_{\alpha \ell(\frac{\lambda}{\alpha}\cdot), \frac{\alpha}{\lambda}\\|\nabla f(x_0)\\|}$ (Note that the initial point for minimizing $g$ is re-scaled to $x_0/\lambda$). From this equation, we can see that $\lambda$ indeed matters. In fact, this equation is very consistent with our step-size choice in Theorem 4.2: $\gamma_{\ell, \\|\nabla f(x_0)\\|}\approx \frac{1}{\ell(\\|\nabla f(x_0)\\|)}$ (which we believe is also worse-case optimal). With this choice, both sides of the equation are the same.
>
> For the case where $\ell(x)=L_0+L_1x$, the step-size is $\frac{1}{L_0+L_1\\|\nabla f(x_0)\\|}$, which does depend on $L_1$. Therefore, the worse-case dynamics do depend on $L_1$ and there is indeed a hierarchy in our function classes. Regarding your point about lack of homogeneity, it is true that when you scale $f$ to $g=\alpha f$, the constant $L_1$ does not change much. However, that only applies to the specific setting where $\ell$ is linear. For example, consider the function $f(x)=1/x$ which is $(3/2, 0, 2)$-smooth per Definition 3 in our paper. After scaling it to $\alpha/x$, it becomes $(3/2, 0, 2/\sqrt{\alpha})$-smooth, which means the constant $L_{3/2}$ changes from $2$ to $2/\sqrt{\alpha}$.
>
> **2. Regarding the second point**, we want to clarify the following several points in your argument
> - (1) The set $\mathcal{S}:=\\{x\in\mathcal{X}, f(x)\le f(x_0)\\}$ is NOT compact. We only assume $f$ is closed (the definition of a closed function is that all of its sub-level sets are closed). So $\mathcal{S}$ is not necessarily bounded (thus not necessarily compact as well). Consider the closed function $f(x)=\exp(-x)$ with domain $\mathbb{R}$ which satisfies all our assumptions. Clearly $\mathcal{S}$ is not compact for this example and the iterates also go to infinity. Since $\mathcal{S}$ is not compact, it is not so straight-forward to bound gradient norm or Hessian within it.
> - (2) The assumption of continuity of Hessian is not necessarily weaker than $\ell$-smoothness, for the latter Hessian may not exist over some points (with zero measure). For example, our example function used to show the lower bound in Theorem 5.3 has some points where Hessian does not exist.
> - (3) However, if assuming $\ell$-smoothness with a sub-quadratic $\ell$, we can indeed bound gradient norm (and thus also Hessian) within it using our Lemma 4.5. However, we want to point out that, Lemma 4.5 is derived based on Proposition 3.3 (the equivalence between $\ell$-smoothness and $(r,\ell)$-smoothness), which is indeed nontrivial and one of our novel contributions. This approach does offer a different way to proving convergence of GD. We were actually aware of this alternative approach and already mentioned it briefly in Section 5.3 (Lines 326--333)
> - (4) However, this approach only works easily for the simple algorithm GD, not for NAG or SGD. Because for NAG, SGD, and potentially other more complicated algorithms, the function values are not really decreasing. For SGD, iterates can easily escape from $\mathcal{S}$ due to its heavy-tailed noise. For NAG, although we can easily bound $f(x_t)$, what we really need is a bound on $f(y_t)$, which turns out to be quite challenging.
>
> **3. Regarding the third point**, although our bounded noise assumption does not apply to linear regression with MSE loss, we think it is the most standard assumption in the optimization literature and is interesting and challenging enough. Noise with bounded variance is actually heavy-tailed and SGD with such noise is much more challenging than deterministic GD. For example, consider the function $f(x)=1/x+1/(1-x)$ with a bounded domain $(0,1)$. With noise, the iterates may easily go outside of the domain. Indeed, if you keep running, it will go outside with probability $1$. So the key is to show that with high probability, it converges before going outside. Also, since it is heavy-tailed, applying a naive union bound does not work and we feel it is necessary to use a stopping time analysis (and optional stopping theorem) which is novel compared with classical one-step analysis.
>
> We thank the reviewer for pointing out the expected smoothness condition. We feel it should be doable and there are actually recent papers studying this condition for $(L_0,L_1)$-smooth function and variance reduction methods (see [Reisizadeh et al., 2023] cited in our paper). However, it is not necessarily weaker than our bounded variance assumption, and we have reservations about whether the former is more interesting or challenging than the latter.
>
> Finally, we thank the reviewer for suggestions regarding the writing, typos, etc., and will update them accordingly in the revision.

---

> > ### Comment · Reviewer_cK5L · 2023-08-21
> > **Response to rebuttal**
> >
> > First of all, I thank the authors for their very detailed answer.
> >
> > 1. I agree with their response, since finally what really matters to tune the parameters is essentially the Lipschitz constant that holds onto the trajectory of the algorithm. And indeed, this reasoning is consistent with their parameter choice of Thm 4.2, which is a good point.
> >
> > 2. I also agree that my reasoning was quick, handy made, and skipping some technical difficulties.
> > I agree that some statements made by the authors like Lemma 4.5 is based on the non trivial Proposition 3.3. Note it does not mean this was necessary and I would be surprised that there is no simpler proof.
> > I agree that analysis for non monotone methods like NAG are less easy to handle. Typically ||\nabla f|| should be replaced by some Lyapunov that works for the given algorithm.
> >
> > Typically, I would expect a proof that would be based on the following:
> > - Under L-smoothness, we know Lyapunov function $V^{L}$ (I make the dependency in L as this is what will change under $\ell$-smoothness). Of course, $V^{L}$ typically depends on the iterate distance to optimum, on the function value and on the gradient norm.
> > - We show some result like $\|\nabla f(x)\|^2 \leq V^L(x)$.  $V^L$ often has a composant $\|\nabla f(x)\|^2$ (when well chosen NAG's Lyapunov has one) which makes the inequality trivial. Otherwise, some more computation may be needed for instance to bound $\|\nabla f(x)\|^2$ by $f(x) - f_\star$ (might be hard in this case).
> > - Then, since the hessian is bounded by a function of the gradient norm, i.e.  by a functiton of the lyapunov, we might be able to bound the hessian, which tunes the algorithm under smoothness (the difficulty lies in the fact that the lyapunov depends on the tuning, which might bring a constraint on the parameter setting that is not so easy to verify) and since the Lyapunov is decreasing assuming smoothness, we keep the same. smoothness later on.
> >
> > Of course, I agree this reasoning is again an intuition that needs more formalization.
> > Also, I acknowledge that, as for now, I do not have a proper better way to propose to solve the problem this paper address. Then, I will not stand against acceptance for the only reason that I think there might be much simpler ways to do it.
> >
> > And since I do not have other reason to reject this paper that addresses a new problem with theoretical guarantees, I will upgrade my score up to weak accept.
> >
> > - I will conclude on the third point. I still think that this assumption on the noise should be banned from optimization, especially in paper that introduce a new class for purpose of generalization, a class that contains quadratics and which stochastic assumption exclude linear regression.
> > As of the difficulty, I indeed thought of non constrained problem when saying that, which only adds up a constant at each step and the residus is used to tune the algorithm, making analysis more than easy. But indeed, in this case, I understand the need for stopping times.
> > So it might indeed be more technical than I claimed, but still not very interesting in my opinion.
> >
> > In any case, this is subjective, and I based my score on the deterministic contribution. I still consider that an expected-smoothness like assumption should be preferred. I don't see it used in the paper authors mentioned. And I would say it is actually a weaker assumption than the one authors used. Indeed expected smoothness is directly implied, in the smooth setting, by the existence of a variance of the noise on the optimal point only. A generalization of this idea in the $\ell$-smooth setting would surely be weaker that assuming existence and uniform bound on the variance of all the gradients.
> >
> > Best regards.

---

### Official Review · Reviewer_MmTX · 2023-07-09

**Soundness:** 4 excellent
**Presentation:** 4 excellent
**Contribution:** 4 excellent
**Rating:** 8
**Confidence:** 4

**Summary:**

This paper generalizes the classic Lipschtiz smooth gradient condition, as well as a recent improvement. The proposed condition is essentially saying the Hessian norm is bounded by a non-decreasing function of the norm of gradient. Using such conditions, the authors proved convergence rates of gradient descent under various settings (convex, strongly convex, non-convex, stochastic). Classic optimal rates are recovered.

**Strengths:**

The paper is well presented. The technical contribution looks original and significant, as it is a generalization of the classic smoothness. Intuitions are shared for proving the convergence.

**Weaknesses:**

Overall, I do not have major concerns.

1. The proof of Theorem 5.1, in particular the split of two time steps in (4) seems delicate and interesting, but also a bit out of blue to me. Is this splitting novel, or did part of it show up in the literature? What motivates such a splitting?
2. Do you have examples of objective functions that show up in machine learning or other applications, such that they satisfy the generalized smoothness, but not the conventional smoothness? This would add much significance to the paper.
3. It was not clear to me when we assume the function to be C^2 and when not. In the introduction, It appears at first sight that only twice differentiable functions are considered. Then I saw definition 2, so it is not true. Then I got confused again at Proposition 3.3: it seems that for l-smooth to hold you need f to be C^2. Did you assume it somewhere?
4. Line 27: “provide a lower bound”. Do you mean on the number of iterations or on the error, or both?
5. Table 1: Why is it called gradient complexity? Is it the number of times the gradient oracle needs to be queried? I thought a more common name is iteration complexity, but maybe I am wrong.
6. Lines 46-48: “if gradients along the trajectory are bounded by a constant G, then the Hessian norms are bounded by the constant ℓ(G).” Why is this “if-then” true? Say we look at a univariate objective function f: R -> R. Its gradient g is also univariate. Are you saying that if g is bounded by a constant G, then |nabla g| is also bounded? This can not be true, since nabla g can be arbitrarily large.
7. Line 77: “condition number” of what?
8. Proposition 3.6: “If” -> “Suppose”



**Questions:**

See above

**Limitations:**

Yes

---

> ### Author Rebuttal · Authors · 2023-08-09
>
> We thank the reviewer for the positive comments! Below we will try to address the questions of the reviewer.
>
> 1. Regarding the proof of Theorem 5.1, this splitting is indeed novel and we are not aware of any previous optimization analysis using similar techniques. Let us briefly talk about the motivation for this approach below. First, it should be natural to consider $\tau$ since our contradiction hypothesis is equivalent to $\tau<\infty$. Then we know that before $\tau$, everything is well bounded, e.g., gradients and Hessian. Then informally speaking, classical analysis directly gives an upper bound of $S_{\text{uc}}$, the area under the curve in Figure 1. However, since the gradient norm at $\tau$ is very large by definition, if we assume there are no abrupt changes in the curve (which is true for a small enough step size), then in some neighborhood around and before $\tau$, the gradient should stay large. Then a lower bound on the size of such a neighborhood essentially gives you a lower bound on $S_{\text{uc}}$, and potentially leads to a contradiction. So we introduce $\tau_{1/2}$ just to rigorously define the neighborhood.
> 2. Regarding examples in machine learning applications, we think the empirical findings in the papers that study $(L_0,L_1)$-smooth functions (e.g. [Zhange et al., 2019] and [Wang et al., 2022]) can also be used to motivate our more general $\ell$-smooth functions. They observe that when you train a language model, along the trajectory of certain optimizers, log(Hessian norm) is roughly a linear function of log(gradient norm), (i.e., Hessian norm is roughly a polynomial function of gradient norm). So we think our $\ell$-smooth function class can better capture this property.
> 3. This is a good question. We NEVER assume twice differentiability in this paper. Definition 1 only assumes twice differentiability **almost everywhere**, which means there may be a measure-zero set where Hessian does not exist. Our Proposition 3.3 is rigorous. When proving the direction from Definition 2 to Definition 1, we rigorously show that any $(r,\ell)$-smooth function defined in Definition 2 is twice differentiable almost everywhere. This uses the Radermacher Theorem (which states that any Lipschitz function is differentiable almost everywhere) and a covering argument (see Lines 509-517 in the appendix).
> 4. Here we mean a lower bound on the iteration or gradient complexity. We will make it clear in the revision.
> 5. Yes, gradient complexity is exactly the number of times the gradient oracle needs to be queried, i.e., the oracle complexity when the oracle is a gradient oracle. For our theorems, it is equivalent to iteration complexity because each time we exactly query the oracle once. We think it is as common as iteration complexity, we use it because existing lower bounds are usually presented in terms of oracle complexities [Carmon et al., 2017, Arjevani et al., 2019].
> 6. This argument is a direct consequence of Definition 1, which bounds Hessian using gradient norm.
> 7. Here we mean the condition number of the objective function. It is defined as $L/\mu$ for $L$-smooth and $\mu$-strongly-convex functions.
> 8. Thank you for pointing out the typo. We will fix it in the revision.

---

> > ### Comment · Reviewer_MmTX · 2023-08-18
> >
> > Thanks for the response! My questions are well addressed. I upgraded my rating.

---

### Decision · Program_Chairs · 2023-09-21

**Decision:**

Accept (spotlight)

**Comment:**

Overview: Relaxed smoothness conditions have been introduced to study the gradient clipping algorithm, and show that clipping in particular allows fixed step-size convergence without smoothness under this relaxed assumption. This paper further generalizes the relaxed smoothness notion used for clipping by bounding the Hessian by any non-decreasing function of the gradient norm.

Then, fixed step-size convergence is shown under the relaxed smoothness assumption in a variety of settings. The key technical part is to show that assuming a large enough initial bound on the gradients and relaxed smoothness, the gradients remain bounded throughout the trajectory regardless of the setting (convex, strongly convex, non-convex, and even stochastic with some necessary caveats). Then, boundedness of the gradients along the trajectory implies the regular smoothness condition, and thus standard analyses can be used to show convergence of gradient descent.

Pros:

- Instead of listing, the positive points are mostly within all the reviews included in the reviewing process. Interesting Problem Setting, Theoretical results, Thorough literature review, Interesting results.

Cons:

- Not many to be honest.

Overall: Based on the initial reviews + discussion with the authors, the paper has only positive comments, and all the comments raised by the reviewers were adequately tackled by the authors.

The only requirement for the authors is to include (if possible and if space allows) any additional discussion that is useful during the rebuttal phase, in order to improve the paper. Please do not include material that is not presented during the rebuttal and material that cannot be checked, unless another round of review is required.